# Population genomics of *Puccinia graminis* f.sp. *tritici* highlights the role of admixture in the origin of virulent wheat rust races

Yuanwen Guo [1], Bliss Betzen[1,6], Andres Salcedo [1,7], Fei He[1,8], Robert L. Bowden [2], John P. Fellers[2], Katherine W. Jordan [1,2], Alina Akhunova [1,3], Mathew N. Rouse[4], Les J. Szabo [4] ✉ & Eduard Akhunov [1,5] ✉

*Puccinia graminis* f.sp. *tritici* (Pgt) causes stem rust disease in wheat that can result in severe yield losses. The factors driving the evolution of its virulence and adaptation remain poorly characterized. We utilize long-read sequencing to develop a haplotype-resolved genome assembly of a U.S. isolate of Pgt. Using Pgt haplotypes as a reference, we characterize the structural variants (SVs) and single nucleotide polymorphisms in a diverse panel of isolates. SVs impact the repertoire of predicted effectors, secreted proteins involved in host-pathogen interaction, and show evidence of purifying selection. By analyzing global and local genomic ancestry we demonstrate that the origin of 8 out of 12 Pgt clades is linked with either somatic hybridization or sexual recombination between the diverged donor populations. Our study shows that SVs and admixture events appear to play an important role in broadening Pgt virulence and the origin of highly virulent races, creating a resource for studying the evolution of Pgt virulence and preventing future epidemic outbreaks.

*Puccinia graminis* f.sp. *tritici* (Pgt) is an obligate biotrophic pathogen causing stem rust disease in wheat. Epidemics of stem rust have been reported worldwide, including in Africa[1], the Middle East[2], North America[3], Asia[2], Europe[4], and other regions[5]. The high levels of variability in virulence (i.e., ability to overcome individual resistance genes) and frequent emergence of races with new virulence combinations have been well-documented among Pgt populations[1,6,7]. However, many factors contributing to the evolution of Pgt virulence remain poorly understood[8].

The first genome-scale diversity analyses performed on a limited number of agricultural pathogens, including pathogens of several cereal crops, such as *Puccinia coronata* f. sp. *avenae*[9], *Zymoseptoria tritici*[8], and *Parastagonospora nodorum*[10], start uncovering the role of variation in a special class of genes, which encodes secreted proteins referred to as effectors, in adaptation to diverse crop genotypes. Fungal effectors are responsible for establishing a compatible interaction with the host by suppressing defense responses[11] and modulating cellular processes[12,13]. The ability to sense these effectors is one of the strategies used by plants to recognize pathogens and trigger the immune response. The sequence divergence or loss-of-function mutations in effector-encoding genes or their complete deletion could render a resistant host unable to recognize a pathogen[14,15]. The

[1]Department of Plant Pathology, Kansas State University, Manhattan, KS, USA. [2]USDA-ARS, Hard Winter Wheat Genetics Research Unit, Manhattan, KS, USA. [3]Integrated Genomics Facility, Kansas State University, Manhattan, KS, USA. [4]Department of Plant Pathology, University of Minnesota & USDA-ARS, Cereal Disease Lab, St. Paul, MN, USA. [5]Wheat Genetics Resource Center, Kansas State University, Manhattan, KS, USA. [6]Present address: USDA-APHIS-PPQ Field Operations, Kansas State University, Manhattan, KS, USA. [7]Present address: Department of Entomology and Plant Pathology, North Carolina State University, Raleigh, NC, USA. [8]Present address: State Key Laboratory of Plant Cell and Chromosome Engineering, Institute of Genetics and Developmental Biology, Chinese Academy of Sciences, Beijing, China. ✉e-mail: szabolesj@gmail.com; eakhunov@ksu.edu

interaction between the host immune system and the pathogen was postulated to diversify the arsenal of effector-encoding genes in pathogen populations[13]. The signatures of diversifying and direct selection detected in the effector-encoding regions from *Parastagonospora nodorum* and *Puccinia coronata* fungi further corroborate the effectors' involvement in adaptation to genetically diverse hosts[9,10]. Thus, processes affecting the pathogen's effector content should play a critical role in the evolution of new virulent strains adapted to resistant crop varieties.

Recent genomic studies revealed the importance of structural variants (SVs) in effector complement evolution[14–18]. SVs are more likely to affect effector-encoding genes and contribute to pathogen evolution and adaptation than other types of variation[19,20]. Inter-species comparisons showed that the location of many of these effectors in the repetitive portions of the genome appears to promote a high rate of evolution[18], which led to the development of the concept of a "two-speed genome"[21,22]. However, SVs in the pathogen populations remain poorly characterized due to the limitations of the short-read-based genome assemblies. The diversity analysis of the agricultural fungal pathogens has been largely focused on small SVs[8–10], or detecting gene presence/absence variation based on the depth of read coverage[10]. Only recently, a reduction in the cost of long-read sequencing allowed for assembling multiple ~40-Mb-size genomes of the wheat pathogen *Zymoseptoria tritici* and characterizing population-scale SVs[23]. Characterization of the scope and distribution of SVs in Pgt populations could help to understand their impact of effector complement and potential role in the origin of virulent strains.

Both somatic hybridization[24] and sexual recombination between diverged isolates are among two other factors that have the potential to diversify effector complements in Pgt and lead to the origin of new virulent strains. The origin of the Ug99 lineage of Pgt, showing a broad virulence spectrum, was associated with somatic hybridization[25]. Likewise, the Pgt isolates collected from an alternate host *Berberis* spp., which is required for the sexual reproduction of Pgt, showed broad virulence and the potential to cause stem rust epidemics[6]. However, the relative contribution of somatic hybridization and sexual recombination to the evolution of new Pgt lineages and virulent strains remains to be investigated.

In this study, the dikaryotic genome of Pgt, which has two nuclei, is sequenced using a combination of long- and short-read sequencing technologies. We use Oxford Nanopore long-read sequence data to construct a chromosome-level haplotype-resolved ~180 Mb genome assembly of the 99KS76A-1 isolate (race RKRQC). This assembly is combined with the previously assembled haplotypes of the Pgt21-0 isolate[25] to use as a pan-genome reference for characterizing genomic diversity in a global panel of Pgt isolates. We analyze the patterns of single nucleotide polymorphism (SNPs) and SVs across the Pgt genome to characterize the genetic composition of and genetic differentiation among regional samples of Pgt isolates with the goal of tracing their origin and identifying adaptive variation. Using comparative analysis of the haplotype-resolved Pgt assemblies, we characterize SVs in the diverse panel of Pgt isolates. Our results indicate that SVs, many of which harbor effector-encoding genes, are under stronger purifying selection than neutral variants in the genome. In combination with the enrichment of effectors in the highly repetitive regions of the Pgt genome, which are prone to generate new SVs at an elevated rate, our findings suggest that SVs contribute to the fast evolution of the effector repertoire in Pgt and likely facilitate adaptation to new host genotypes from distinct geographic regions. By studying global and local ancestry in the Pgt panel, we show that the origin of 8 out of 12 phylogenetic clades is associated with inter-lineage admixture in the populations undergoing either somatic hybridization or sexual recombination. The virulence of the Pgt panel is characterized by a diverse set of wheat lines carrying distinct stem rust resistance genes. The increase in virulence observed in the group of admixed isolates suggests that hybridization between distinct lineages likely play an important role in the evolution of Pgt virulence.

## Results

### 99KS76A-1 genome assembly and annotation

The 99KS76A-1 isolate was collected in Kansas, USA, in 1999 and was used for identifying one of the first Pgt effectors recognized by a wheat resistance gene[14]. It was formerly identified as race RKQQC but was recently re-phenotyped and reclassified as race RKRQC. To create a haplotype-resolved assembly of the 99KS76A-1 genome, we used the combination of the Hi-C technique, long-read Oxford Nanopore (OXN) sequencing with short-read Illumina sequencing (2 × 300 bp). A two-step pipeline was utilized for scaffolding the contigs generated using Canu assembler[26] as described in the Methods. The first step was based on a bin-assignment strategy[25] (Supplementary Tables 1, 2), which resulted in 120 contigs assigned to 33 bins with a total contig length of ~150 Mb. Step two was based on the ALLHiC assembler[27,28] that utilized Hi-C reads to create a chromosome-level assembly of the 99KS76A-1 genome, which resulted in 36 chromosomes with a total length of ~161 Mbp (Fig. 1a). The assembled chromosomes were assigned to haplotypes from distinct nuclei using Hi-C data, as previously described[25,29]. Out of 36 chromosomes, 34 could be assigned to distinct haplotypes. The 99KS76A-1 haplotypes were designated as "E" and "F" (Supplementary Fig. 1), which showed 97.1 and 96.8% similarity, respectively, to haplotype "A" of Ug99, and 96.7 and 97.4% similarity, respectively, to haplotype "C" of Ug99 (Fig. 1b). Our chromosome assemblies were largely collinear with the Pgt21-0 genome[25] (Supplementary Fig. 2 and Fig. 1c), except for chromosome 8. The comparative analysis of homologous chromosomes suggests that one of the haplotypes of chromosome 8 in Pgt21-0 is likely rearranged relative to the 99KS76A-1 genome (Supplementary Fig. 2 and Fig. 1d). Together with the 985 contigs not assigned into chromosomes, the final assembly of 99KS76A-1 includes 1021 scaffolds/contigs with the total length of 181 Mb. The N50 of assembled scaffolds was 4.5 Mb, with the largest scaffolds being 7.0 Mb (Supplementary Table 2).

The completeness of the 99KS76A-1 assembly was evaluated using BUSCO[30]. The proportion of fragmented and missing genes was lower in the 99KS76A-1 genome assembly compared to that of the Pgt21-0 and Ug99 genomes[25] (Fig. 1e and Supplementary Table 3). The proportion of complete single-copy genes was nearly two times higher in our assembly compared to the previously reported Pgt21-0 and Ug99 assemblies, which is consistent with the reduced number of duplicated genes in BUSCO analyses for individual haplotypes E and F (Supplementary Table 3). There was a slight decrease in the number of duplicated genes in 99KS76A-1 compared to Pgt21-0 and Ug99 assemblies (Supplementary Table 3). This difference could possibly be explained by the lower divergence between some regions of the two 99KS76A-1 haplotypes, which does not allow for assembling fully resolved scaffolds. In this case, the single-copy genes detected by BUSCO in the assembly are potentially located within the regions where reads from both haplotypes, due to high similarity, were collapsed together by an assembler. However, if some reads still overlap with the sites that distinguish one haplotype from another, they could be detected as heterozygous SNPs in the alignment of the 99KS76A-1 reads to the genome assembly. The 346 single-copy genes detected by BUSCO in the assembly overlapped with 5412 SNPs segregating in the entire Pgt diversity panel. These single-copy genes were randomly distributed across the genome with an average interval of 0.4 Mb. Only 20 of these sites (0.3 %) were heterozygous in the 99KS76A-1 isolate, indicating that most of these genes are either indeed single copy in our assembly, or they have nearly identical haplotypes between the homologous chromosomes from distinct nuclei.

The alternative haplotypes in the 99KS76A-1 genome assembly were validated by aligning annotated genes among scaffolds. All

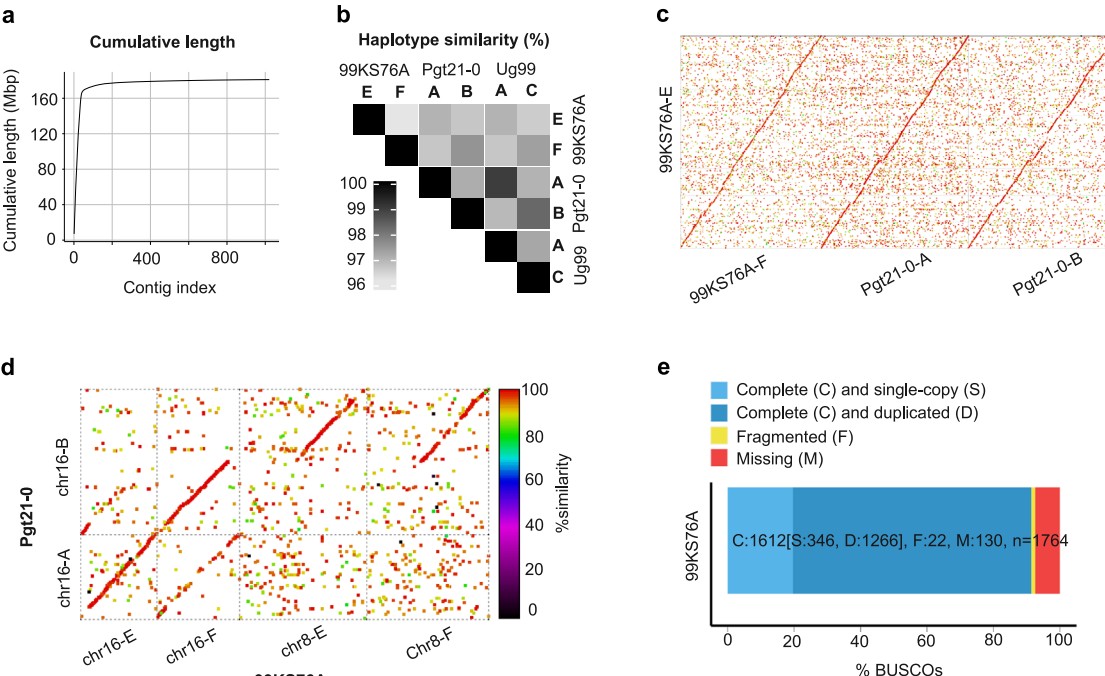

**Fig. 1 | Summary of the 99KS76A-1 genome assembly. a** Cumulative length versus the size-ranked scaffolds. **b** Pairwise comparison of sequence similarity among Pgt haplotypes A, B, C, E, and F. **c** Whole-genome sequence comparison of haplotype E from 99KS76A-1 with haplotype F from 99KS76A-1, and haplotypes A and B from Pgt21-0. **d** Sequence alignment of homologous chromosomes in 99KS76A and Pgt21-0 suggests that one of the haplotypes on chromosome 8 in Pgt21-0 is rearranged compared to 99KS76A-1. **e** Assessment of the 99KS76A-1 assembly completeness using Benchmarking Universal Single-Copy Orthologs (BUSCO).

36 scaffolds identified in our previous analyses were shown to represent distinct haplotypes, with each homologous pair sharing at least 200 genes. Based on BUSCO analysis, each haplotype assembly carried 1,266 duplicated genes suggesting that most of the 99KS76A-1 genome was haplotype-resolved (Fig. 1e). These results are consistent with alignments between the 99KS76A-1 genome scaffolds (Supplementary Fig. 3), which indicates that assembly is haplotype-resolved. Evidence-based annotation using previously developed resources[14,25,31,32] predicted 22,974 genes with an average CDS length of 929 bp (Supplementary Table 4). Among them, 3952 effector-encoding genes were predicted with an average CDS length of 957 bp (Supplementary Table 5 and Supplementary Data 1).

### Genetic diversity and linkage disequilibrium in the Pgt panel

We investigated the patterns of genomic diversity in a panel of 77 diverse Pgt isolates collected in the United States and other parts of the world, including a group of isolates from the Ug99 lineage (Supplementary Data 2). The panel includes isolates collected from 1959 to 2014 in Europe, the Middle East, Africa, Asia, and North America and represents entries assigned to eight previously identified Pgt clades[33–35]. In total, we generated 1,471,824,965 raw read pairs for the entire panel, with an average of 18,869,551 read pairs per isolate. Because structural variation (SV) between the two haplotypes of 99KS76A-1 might affect the variant discovery rate, read mapping was done to each of the E and F haplotypes separately. On average, about 40% of read pairs were uniquely mapped to individual haplotypes providing ~25X coverage per haploid genome. SVs between the haplotypes were identified by aligning haplotype F against haplotype E[36] (see Methods). After filtering for read depth and allele frequency (see Methods), we identified 2,208,026 SNPs and small-scale indels in haplotype E. After adding variants discovered in the scaffolds missing in one of the haplotypes, we obtained 2,325,518 variants (2,197,381 SNPs and 128,137 indels). Using this data, the average estimates of genetic diversity ($\pi$) in the Pgt panel were $4.6 \times 10^{-3}$ for the introns, $3.4 \times 10^{-3}$ for the genic regions, and $3.0 \times 10^{-3}$ for exons.

The potential functional effects of 2,229,058 variants were predicted using the SnpEff[97] (Fig. 2a and Supplementary Tables 6–8). Only a small proportion of variant effects were high impact (0.17%) or nonsense (0.59%) mutations (Supplementary Tables 6, 7). We identified 187,085 nonsynonymous (3.4%), and 272,321 synonymous mutations (4.9%) (Supplementary Tables 9, 10). The site frequency spectrum (SFS) of SNPs with strong effects resulting in stop codons and splice site disruptions was shifted towards rare variants, which was different from the frequency spectrum of nonsynonymous and synonymous variants ($p$ values were 0.05 and 0.02, respectively) (Fig. 2b, c). These differences in SFS between various SNP types remained detectable in even more genetically homogeneous geographic subpopulations, after the Pgt panel was split into the US and non-US groups (Supplementary Fig. 4). These results are suggestive of purifying selection acting against strong effect mutations in Pgt.

Recombination and demographic aspects of population history (population size changes, admixture, population structure) affect linkage disequilibrium (LD), making it a useful summary of diversity patterns in the populations. We compared LD decay in the United States and non-US samples of Pgt isolates by fitting the LD decay model described by ref. 38. This comparison revealed a slow rate of LD decay in both samples. However, the rate of LD decay in the United States was slower than in the non-US samples, where LD reduced from $r^2$ ~ 0.5 to $r^2$ ~ 0.25 within 1.8 Mb compared to 4.5 Mb in the United States (Fig. 2d). Among factors explaining such differences in LD decay could be either change in effective population size due to population bottleneck or reduced frequency of inter-lineage recombination in the US linked with the eradication of barberry[39], which is required for Pgt to accomplish sexual cycle.

### Effector-encoding genes are enriched in the genomic regions subjected to selection

Previous studies showed that fungal effectors evolve faster than other genes[10], likely in response to high selection pressure associated with the need to evade recognition by the host's resistance genes.

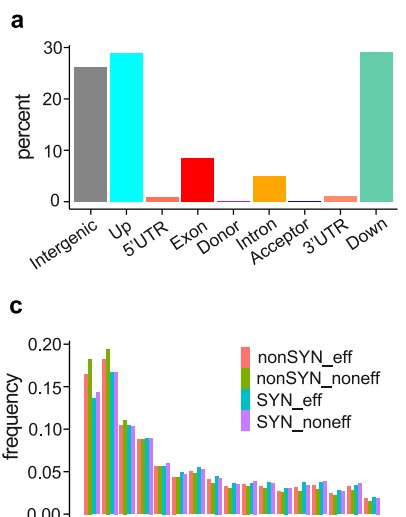

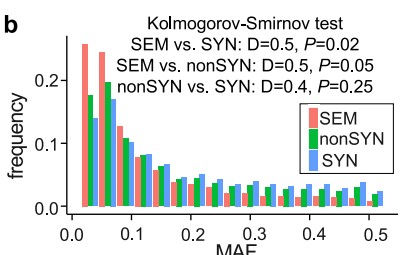

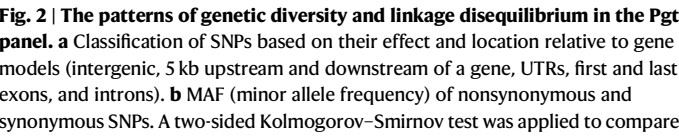

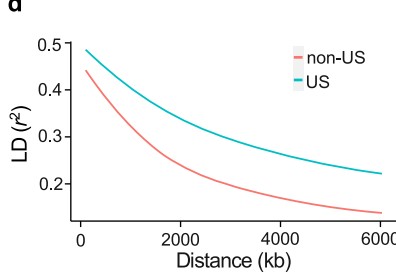

**Fig. 2 | The patterns of genetic diversity and linkage disequilibrium in the Pgt panel. a** Classification of SNPs based on their effect and location relative to gene models (intergenic, 5 kb upstream and downstream of a gene, UTRs, first and last exons, and introns). **b** MAF (minor allele frequency) of nonsynonymous and synonymous SNPs. A two-sided Kolmogorov–Smirnov test was applied to compare MAF spectra between these two types of SNPs. SEM strong effect mutations include stop codon gain and splice site disruption. **c** MAF of synonymous and nonsynonymous SNPs located within predicted effectors and non-effectors. **d** The rate of LD decay in the Pgt samples from the US (15 isolates) and outside of the US (22 isolates) was modeled as a function of the physical distance between SNPs.

While in the entire Pgt panel, effector-encoding genes show a significantly lower difference in the levels of diversity ($\pi = 3.2 \times 10^{-3}$) compared to non-effector-encoding genes ($\pi = 3.4 \times 10^{-3}$) (Fig. 3a), the relative ratio of nonsynonymous to synonymous mutations (pN/pS) was substantially higher in effector-encoding genes (Fisher's exact test, $p = 3.3\text{e-}16$) (Supplementary Table 11 and Fig. 3b). This result is consistent with either the relaxed or positive selection for nonsynonymous mutations. However, because the relative ratio of mutations creating stop codons and splice site disruptions to synonymous mutations was similar between the effector-encoding genes (0.015) and other genes (0.013), it is likely that the accumulation of nonsynonymous mutations is driven by positive selection rather than by the relaxation of selection, which would lead to accumulation of mutations rendering effectors non-functional.

To better understand the role of effectors in adaptation to both region-specific host genotypes and local environments, we calculate effector enrichment within the genomic regions showing the signatures of selection. We performed selective sweep scans across the Pgt genome by comparing a sample of the US isolates with the reference population composed of isolates from Africa, Europe, and Asia. These two groups included only those Pgt isolates that show low levels of inter-group admixture at the optimal $K = 6$ (Supplementary Table 12). To reduce the proportion of falsely detected selective sweeps, we conducted selection scans using two different methods, the cross-population composite likelihood ratio test (XP-CLR v1.0)[40] and SweeD[41], and detected common outliers in both scans. Both methods were previously shown to be robust to variation in demographic models and recombination rate[40,41]. However, to reduce the chance of demographic events affecting the results of selection scans, we investigated the selection scans' test statistics thresholds by conducting selection scans in the simulated datasets considering the demography of the two Pgt samples (see Methods and Supplementary Material).

The analyses of simulated data suggested that empirical thresholds defined in both scans as the 95th percentiles of test statistic distribution are conservative and should not increase the rate of falsely detected selective sweeps. We identified 1602 genomic outlier regions shared by both the XP-CLR and SweeD scans, and

among them, 159 genomic regions overlapping with 120 effectors (Supplementary Table 13). The number of selected regions overlapping with effectors (159) is nearly two times higher than the average number of randomly selected regions of equal size overlapping with effectors (80) (Fig. 3c). These results suggest that region-specific selection in the US population preferentially targeted effector-encoding genes. Consistent with this hypothesis, previously identified AvrSr35[14] and AvrSr50[15] were located within 0.8 and 9.2 kb from the nearest outlier regions (Fig. 3d). The effect of selection on the candidate effector genes was also evident from the (1) statistically significant reduction of Tajima's D and $\pi$ in the United States compared to the non-US samples and (2) significant increase in interpopulation genetic differentiation $F_{ST}$ compared to the effector-encoding and other genes located outside of the selective sweep regions (Fig. 3e–g).

## Structural variants (SVs) show evidence of selection in the Pgt panel

The impact of SV on the Pgt effector content remains poorly characterized. Here, we used MUMmer[36], Assemblytics[42], and custom scripts for detecting four types of SVs (insertion, deletion, contraction, and expansion) among the four assembled haplotypes of the 99KS76A-1 and Pgt21-0 isolates. A total of 20,576 SVs affecting 79 Mbp of the genome were detected. The size of each SV ranged from 50 bp to 1.3 Mbp, with most SVs being smaller than 5 kbp (Fig. 4a). In the 99KS76A-1 genome, detected SVs overlapped with 5824 genes and 947 effectors (Supplementary Data 3), indicating that SVs could affect the Pgt isolates' effector content.

Some of the large-scale SVs affected multiple effectors. For example, a 657,961 bp-long contraction on chromosome 12-F causes the loss of 18 effectors (Fig. 4b). In addition to these 18 effectors, a total of 79 effectors (35%) on chromosome 12 of 99KS76A-1 genome were affected by SVs (Fig. 4b). There were additionally 27 effectors (12%) present in only one copy of the four homologous haplotypes from 99KS76A-1 and Pgt21-0 but were not captured by our SV detection process. This result might be associated with the low levels of sequence collinearity between the regions of homologous chromosomes that may complicate the detection of SVs.

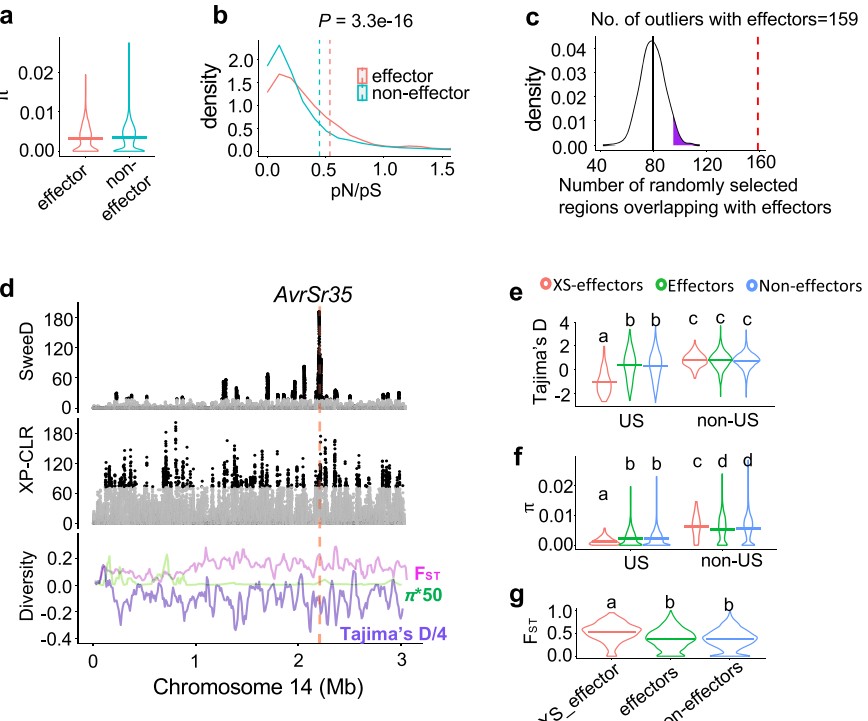

**Fig. 3 | The regions of the Pgt genome showing evidence of adaptive selection are enriched for predicted effector-encoding genes and overlap with aviru-lence factors. a** Violin plots of genetic diversity $\pi$ for effector/non-effectors (Two-sided Mann–Whitney $U$-test $p$ value = 0.016). **b** Density plot of gene-based pN/pS ratio calculated for effector and non-effector-encoding genes (Two-sided Mann–Whitney $U$-test $p$ value = 3.3e-16). The mean and SEM (standard error of the mean) for effectors is 0.54 and 0.05, mean and SEM for non-effectors is 0.45 and 0.024. Outliers of pN/pS greater than 1.5 are not displayed. **c** Density plot showing the number of genomic windows overlapping with effectors in the random sample of genomic windows. Random sampling was performed 1000 times. The 95th percentile of the density plot is highlighted in purple. The number of selective sweep outlier windows overlapping with effector-encoding genes (159) is shown by a vertical dashed line. A selective sweep is performed using 15 isolates from the US

and 22 isolates from Africa, Europe, and Asia. **d** The distribution of SweeD and XP-CLR test statistics (top plots, outlier regions exceeding the 95th percentile threshold are represented with black dots, while others are gray dots), $\pi$, $F_{ST}$, Tajima's D (bottom plot) for the *AvrSr35* region on chromosome 14. **e**–**g** Distribution of Tajima's D, $\pi$, and $F_{ST}$ among the three sets of genes in the Pgt samples used for the XP-CLR and SweeD selection scans, 15 isolates from the United States and 22 isolates outside of the United States. Three groups of genes include effectors within the selective sweep regions (XS-effectors), effectors, and non-effectors outside of the selective sweep regions. Groups with different letters correspond to statistically significant differences based on the results Kruskal–Wallis (KW) test ($p$ value = 2.2e-16 in all comparisons presented in 3**e**–**g**) followed by the Dunn's test with the Benjamini–Hochberg correction.

Further, we compared the relative distribution of SVs, TEs, and effector-encoding genes across the genome. The proportion of the Pgt genome affected by SV was directly related to the TE content ($r^2 = 0.67$) (Fig. 4c), consistent with the TE-mediated mechanisms of SV origin[43]. The parts of the Pgt genome subjected to structural variation tended to be enriched for effector-encoding genes (Fig. 4c), indicating that TEs could contribute to the evolution of Pgt effector repertoire and thereby affect Pgt virulence.

To characterize the genomic distribution and frequency of SVs in the Pgt panel, we applied k-mer-based analysis to the raw Illumina sequencing data generated for the Pgt isolates (described in Methods). A diagnostic set of 31 nucleotide-long k-mers tagging both presence and absence alleles at each of the 11,517 SV sites (insertion and deletion) (Supplementary Fig. 5) was used to call SVs the Pgt panel. The NJ tree based on SV genotypes in the panel (Supplementary Fig. 6) was in good agreement with the tree constructed using the SNPs (Fig. 5a). A total of 173 SVs genotyped in the Pgt panel overlap with the 190 predicted effector-encoding genes (Supplementary Data 4).

To further test whether the SVs affecting effectors are under selection, we compared the allele frequency spectra between synonymous SNPs and SVs. For this purpose, SNPs between the four haplotypes of 99KS76A-1 and Pgt21-0 were detected with MUMmer[36] show-snps, generating an SNP set which is comparable with the SV dataset. In these datasets, SV frequency was shifted towards

more rare variants compared to the frequency of synonymous SNPs, suggesting that SVs are likely under purifying selection (Fig. 4d). To assess the potential adaptive value of SVs, we calculated single-locus pair-wise estimates of $F_{ST}$ between the Pgt samples from different geographic regions (Fig. 4e). Although, the average of $F_{ST}$ for SVs (0.2) was lower than that for SNPs (0.3), there were SVs showing high levels of genetic differentiation between the United States and non-US Pgt populations. Among 38 highly differentiated SVs with $F_{ST}$ values above 95th percentile ($F_{ST} = 0.7$) of genome-wide $F_{ST}$ distribution, we identified two SVs (insertion sites at chr5-E: 4106919-4106921 and chr14-E: 562512-562513) overlapping with the effector-encoding genes (asmbl_25804.p1, asmbl_3632.p1). This result, combined with the enrichment of effector-encoding genes within the selective sweeps regions (Fig. 3c, g), suggests that variation in the effector content induced by SVs could potentially be adaptive.

Two previously characterized effectors, *AvrSr35*[14] and *AvrSr50*[15] have also been detected in this SV analysis and labeled as asmbl_4405.p1 and asmbl_4399.p1 on chromosome 14 group 2 (Supplementary Data 3, 4). The geographic distribution of SV alleles shows that the 408-bp-long AvrSr35 MITE insertion is more frequent in the US, whereas the AvrSr50 with the 26,789 bp insertion is more common in Africa (Fig. 4f, g). Combined with the results of selection scans (Fig. 3f) showing evidence of the selective sweep at the *AvrSr35* locus, the association of SV alleles with specific geographical regions may

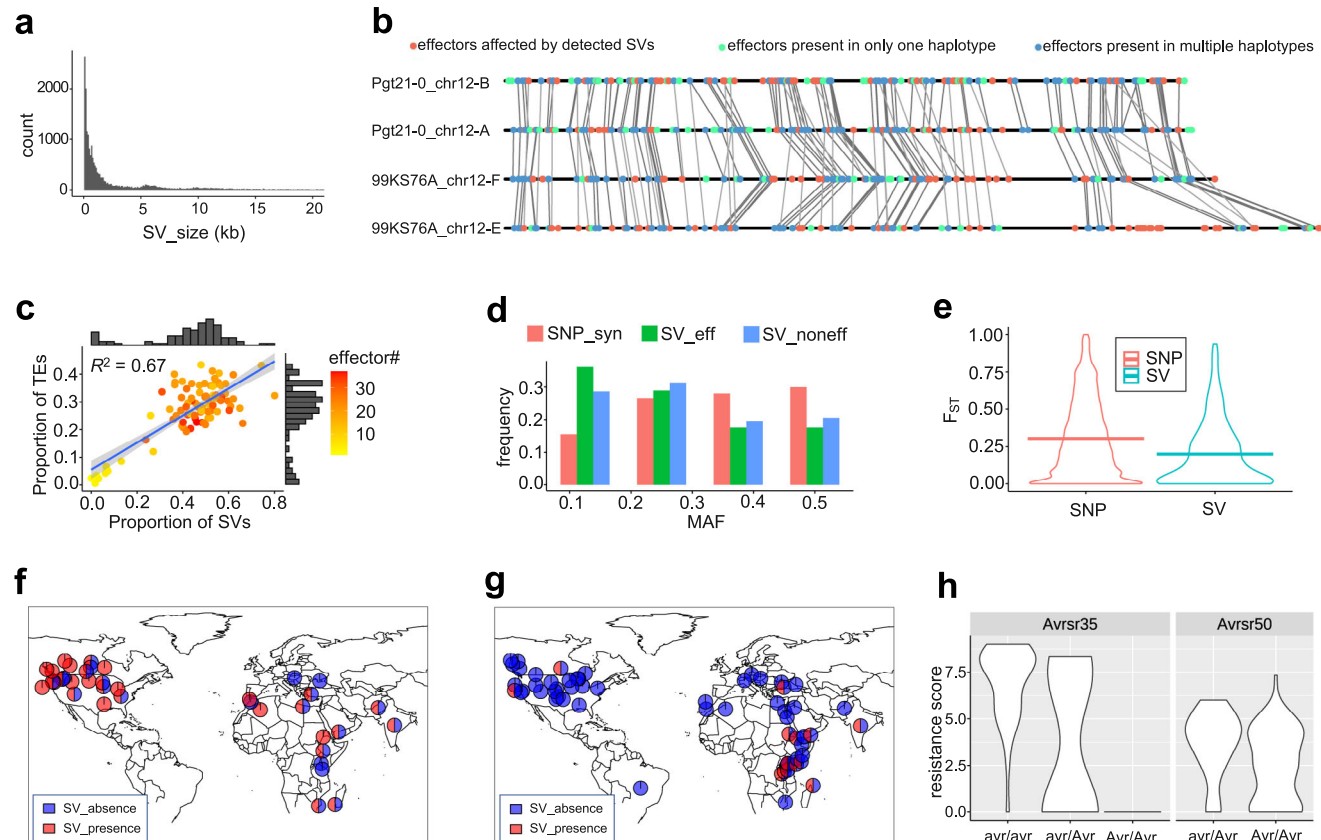

**Fig. 4 | The SV size, SV distribution, and pairwise genetic differentiation at SV sites between the groups of Pgt isolates from different geographic regions.** **a** SV size distribution. Outliers of SV size greater than 20 kbp are not displayed. **b** The panel shows the synteny map of effector-encoding genes between the four homologous chromosomes from the 99KS76A-1 and Pgt21-0 isolates. Effectors affected by SVs are represented with coral dots, green dots show effectors that are present in only one haplotype, and blue dots show the remaining effectors. The effector-encoding genes collinear between the Pgt haplotypes are connected using gray lines. **c** Relationship between the TE content, structural variation, and predicted effector content. The proportion of TEs and SVs within each non-overlapping 1 Mb window was calculated across the 99KS76A-1 genome. The best-fit regression line (blue) and standard error intervals (light blue) are shown on the graph. Those **d** MAF spectrum of SVs (insertions and deletions) affecting effectors, non-effectors, compared with synonymous SNPs. **e** Pairwise estimates of $F_{ST}$ for SVs and SNPs between Pgt isolates from the United States and outside of the United States. **f**, **g** The geographic distribution of the *AvrSr35* (**f**) and *AvrSr50* (**g**) alleles with and without SV. **h** The SV in both *Avr* genes leads to increased virulence on wheat lines carrying the *Sr35* (left side of panel **h**) and *Sr50* (right side of panel **h**) resistance genes.

result from adaptation to local host genotypes. The SV in *AvrSr35* has an impact on the isolates' virulence on *Sr35* (Fig. 4h), consistent with the previously published results[14,15].

## Population structure analyses of Pgt isolates

We used whole-genome diversity data to investigate the genetic relationships among the Pgt isolates using ADMIXTURE[44]. The mean and variance of cross-validation error rate in ADMIXTURE suggest that the most optimal value of K in the Pgt panel was close to 6 (Supplementary Fig. 7). We compared the results of ADMIXTURE with the previously identified phylogenetic clades of Pgt isolates[33,34,45], and a new clade IX that includes a group of isolates related to Pgt21-0 (Fig. 5a, b).

At K = 2, clades III, IV, and VII, including isolates from Europe, Africa, and South America, are clearly separated from the North American clade VI-C. At K = 8, isolates from clades III and IV split into two distinct groups with only small fractions of their genomes sharing membership in the same cluster (Fig. 5a, b). Two isolates from clade VII showed an admixed origin at K ranging from 3 to 7, with significant proportions of their ancestry shared with isolates from clades VI-A, III, and IV. The isolates from clade VI-C were largely assigned to the same cluster for K ranging from 3 to 7, and only a small fraction of their genomes were assigned to other clusters at K values of 8 and 9 (Fig. 5a, b). The remaining clades I, II, and VI-A, comprising isolates from Africa and North America, showed varying proportions of their genomes

assigned to distinct clusters for all values of K ranging from 2 to 9 (Fig. 5a, b and Supplementary Figs. 8 and 9).

At K = 7, clade I, which includes some of the most virulent isolates of Pgt from the Ug99 group capable of causing severe rust epidemics[1,45], showed mixed ancestry with membership in two clusters (Fig. 5a, b). One of these clusters shares ancestry with isolate Pgt21-0, which was recently shown to be a tentative donor of one of the Ug99 haplotypes[25]. The second cluster shares membership with a group of isolates from cluster II (Fig. 5a, b), suggesting that this cluster or closely related isolates could be the likely source of the second Ug99 haplotype, which so far remained unknown.

To ensure that ADMIXTURE analysis was not affected by the inclusion of closely related Pgt isolates, we repeated the analysis using a reduced set of individuals, including only one isolate from each of the previously defined asexual lineages[33–35,46] (Supplementary Fig. 10). The ADMIXTURE analysis with the optimal value of K = 4 for this set showed that most isolates showing mixed ancestry also showed mixed ancestry in the analysis of the full dataset at K = 6.

The clustering of Pgt isolates based on the virulence profiles on a set of *Sr* genes and genetic divergence (Supplementary Fig. 11) were largely consistent. Most Pgt isolates from the same geographic regions tended to show similar virulence profiles, except for Asian isolates, which fell into distinct clusters. This outcome is likely associated either with the admixed origin of Asian isolates that likely resulted in new

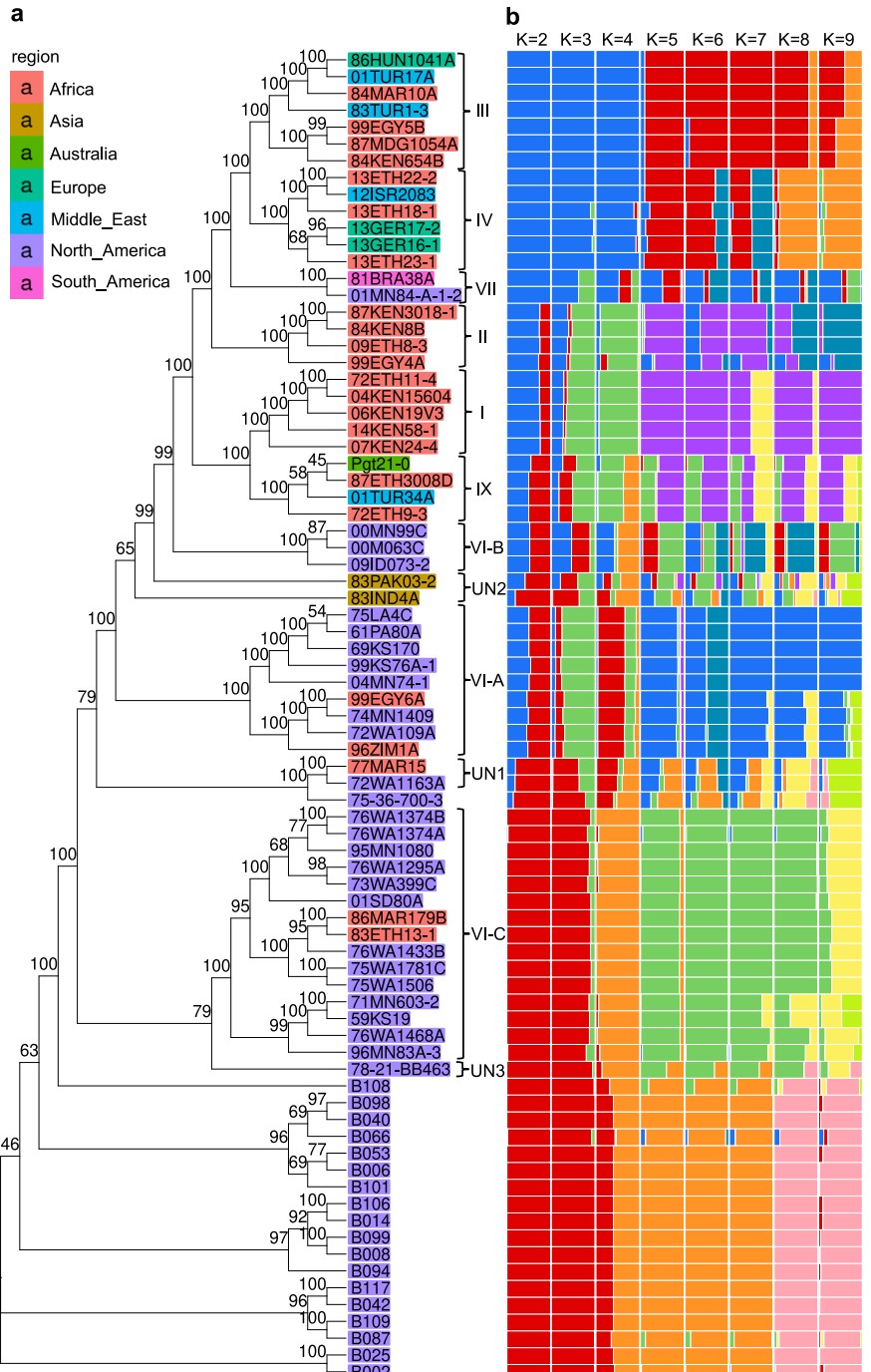

**Fig. 5 | The genetic relationship and population structure of isolates in the Pgt panel. a** The neighbor-joining tree was constructed based on the genetic distance estimated using SNPs. The tree branch lengths are scaled to have the same length from the root to the tips and do not reflect the actual mutation rate along the branches. The bootstrap support values for the phylogenetic tree of the Pgt panel are shown above the branches. The previously used clade designations[34,35] are shown as Roman numerals. Four smaller groups of isolates (1–3 isolates/group) that were previously unassigned to clades were given temporary labels UN1-UN3. **b** The population structure was analyzed using ADMIXTURE for K values ranging from 2 to 9. The average estimates of ancestry proportions in K populations were based on ten independent runs of ADMIXTURE.

combinations of virulence factors or with the accumulation of mutations in avirulence genes within this group of isolates.

**Fine-scale local ancestry analyses of Pgt lineages**

The relative contribution of sexual recombination and somatic hybridization to the origin of Pgt lineages, often showing distinct virulence profiles, remains unknown. To investigate which of these two factors accompanied admixture events and detect the sources of ancestral haplotypes underlying the origin of main Pgt clades, we performed fine-scale analyses of local ancestry along the genome in the entire Pgt panel using MOSAIC[47]. We inferred the local ancestry of isolates derived from a cross between isolates CDL 78-21-BB463 and CDL 775-36-700-3 using $F_2$ isolates as targets and two parental isolates as admixing sources. As expected, the local ancestry along chromosomes clearly shows a mosaic of source haplotypes, consistent with recombination between parental isolates, with a high level of correlation ($r^2 = 0.88$) between inferred and true ancestry (Fig. 6a).

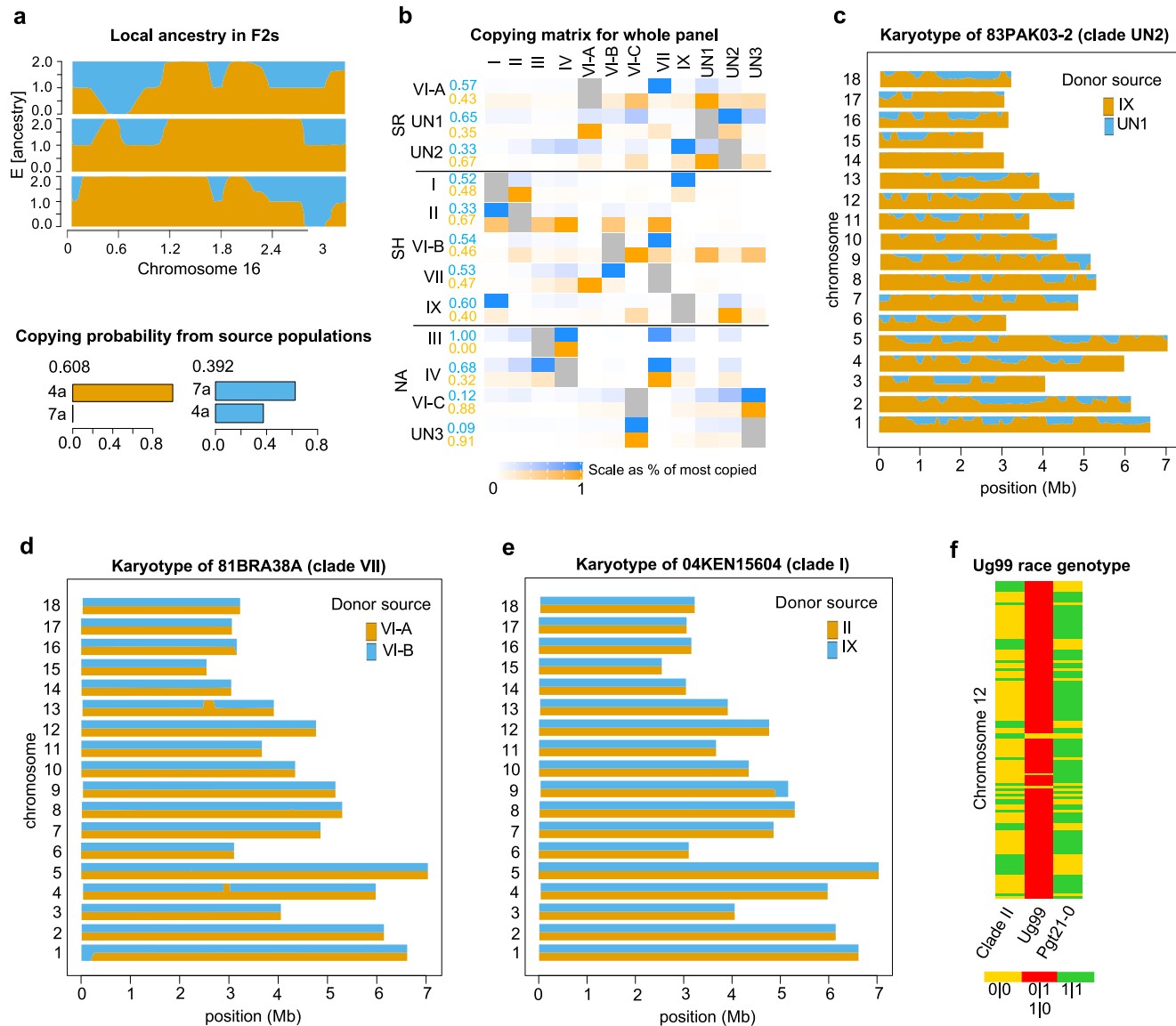

**Fig. 6 | Analysis of local ancestry using MOSAIC. a** Local ancestry of three $F_2$ isolates along chromosome 16 (top panel) expressed as the probability of copying (E[ancestry]) from the two parental isolates 4 A and 7 A (bottom panel). The bottom panel's x-axis shows the copying probabilities of two ancestries from isolates 4 A and 7 A. The proportions of ancestry in the two-way admixture model in the sample of three isolates is shown on the top (0.608 and 0.392). **b** Copying matrix for the clades in the Pgt panel modeled as two-way admixture. Row names show clades used as target panels, and columns show reference panels. The proportions are scaled as % of the most copied within rows. The numbers listed on the left are the averaged copying proportions across the reference panels in a two-way admixture model. Based on the analysis of the local ancestry of individual Pgt isolates, we inferred whether the origin of a clade is linked with sexual recombination (SR), somatic hybridization (SH), or lack of admixture (non-admixed - NA). **c** The genome-wide estimates of local ancestry in the 83PAK03-2 isolate from clade UN2. **d** The genome-wide estimates of copying proportions in two-way admixture for clade VII, which is ancestral to isolate 13ETH18-1. **e** The genome-wide estimates of Ug99 ancestry in two groups of isolates from clades II and IX (includes Pgt21-0). **f** Genotype of Ug99 in comparison with the genotypes of isolates from two sources, a group isolates from clades II and IX. SNPs used in the analysis showed homozygous genotype calls (0|0 and 1|1) in the source isolates and heterozygosity (0|1 or 1|0) in Ug99.

For analyzing admixture between the populations of isolates in the Pgt panel, we used previously defined phylogenetic clades I, II, III, VI-A, VI-B, VI-C, VII[33–35,45,48] (Fig. 5a and Supplementary Data 5). Several groups of isolates with three or less samples that cluster separately from the known clades were temporarily labeled UN1, UN2, and UN3. The group of four isolates, including Pgt21-0, one of the donors of the Ug99 Pgt race, was named Clade IX. Comparison of two-, three-, and four-way admixture models indicate that all admixture events are best explained by a two-way model. By applying the two-way admixture model to 12 Pgt clades, we showed that eight clades (I, II, VI-A, VII, IX, VI-B, UN1, and UN2) show evidence of admixture between the pairs of distinct donor sources (Fig. 6b and Supplementary Table 14). The

MOSAIC results were, in general, consistent with the results of ADMIXTURE analysis at optimal $K = 6$ (Fig. 5b).

To further characterize admixture events, we examined the distribution of local ancestry along the chromosomes for individual isolates from each of these eight admixed clades. MOSAIC fitted clade UN2 (including two Asia isolates) as a two-way admixture ($r^2 = 0.82$) between Clades IX and UN1 (Fig. 6b, c and Supplementary Fig. 12). The local ancestry along the chromosomes of isolates from this clade represents a mosaic of contiguous chunks of genome inherited from these two donor populations (Fig. 6c and Supplementary Fig. 12). This result is consistent with sexual recombination between the ancestral haplotypes from Clades IX and UN1 contributing to the origin of

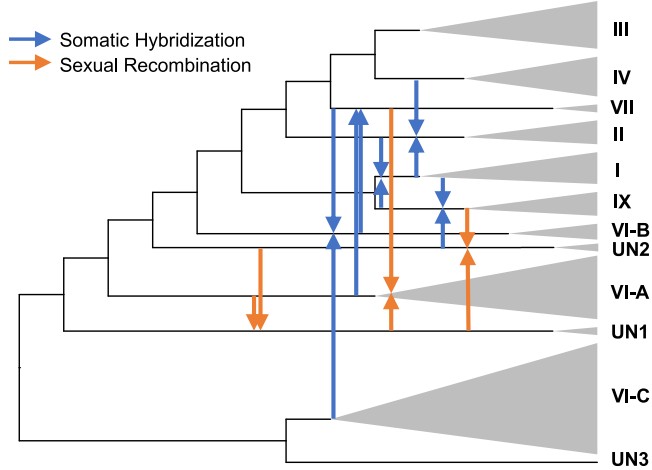

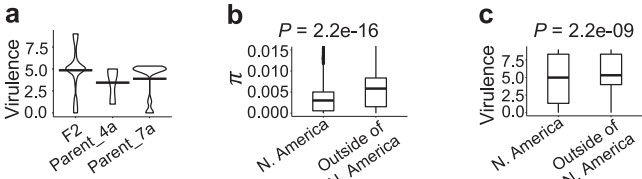

**Fig. 7 | Admixture events between clades in the Pgt panel.** For each of the eight admixed clades identified by MOSAIC, its two distinct donor sources are shown with arrows. The admixture events accompanied by somatic hybridization or sexual recombination between donor populations are shown with different colors. The reciprocal admixture between the pairs of clades suggests that they share a haplotype with similar ancestry.

**Fig. 8 | Relationship between Pgt virulence, admixture, and sexual recombination. a** Violin plot showed increased virulence of the $F_2$ Pgt progeny on disease resistance genes $Sr7b$, $Sr11$, $Sr6$, $Sr9b$, $Sr30$, and $Sr24$ compared to two parental isolates (4a or CDL 78-21-BB463, race BFBJC; 7a or CDL 75-36-700-3, race SCCLC). **b** Genetic diversity ($\pi$) was lower in the sample of North American Pgt isolates ($n = 15$) than in the sample of isolates outside of North America ($n = 22$). (Two-sided Mann−Whitney $U$-test $p$ value = 2.2e-16). Outliers of $\pi$ greater than 0.015 are not displayed. **c** Virulence was reduced in the isolates from North America ($n = 15$) compared to the remaining population ($n = 22$) (Two-sided Mann−Whitney $U$-test $p$ value = 2.15e-09). Box plots show the median and interquartile ranges (IQR). The end of the top line is the maximum or the third quartile (Q) + 1.5 × IQR. The end of the bottom line denotes either the minimum or the first Q − 1.5 × IQR. The dots are either more than the third Q + 1.5 × IQR or less than the first Q − 1.5 × IQR.

83PAK03-2 and 83IND4A. Likewise, Clade VI-A ($r^2 = 0.87$) and UN1 ($r^2 = 0.85$) could be modeled as a two-way admixture between Clades UN1 and VII, and between Clades VI-A and UN2 (Supplementary Figs. 14 and 15). The chromosomal distribution of local ancestry in individual isolates from clades VI-A and UN1 was also consistent with sexual recombination between the source populations (Supplementary Figs. 13–15).

The 2013–2014 Pgt outbreak in Ethiopia, leading to nearly 100% yield loss in the fields planted with a major local wheat cultivar, was attributed to a new race TKTTF (isolate 13ETH18-1 collected in 2013)[33]. On the phylogenetic tree, isolate 13ETH18-1 clustered with isolates from clade IV (Fig. 5a). Inference of local ancestry using MOSAIC suggested non-admixed origin of clade IV, with the majority of the isolates' genomes sharing ancestry with clade VII (Fig. 6b). However, earlier study suggests that the group of isolates from clade IV collected in 2013 could be derived from sexually recombining Pgt population[34]. It is possible that our inability to detect recombined haplotypes in the MOSAIC analyses of clade IV could be associated with the high levels of similarity among recombined haplotypes. The analysis of isolates from clade VII using MOSAIC detected evidence for two-way admixture ($r^2 = 0.88$) between clades VI-A and VI-B (Fig. 6d and Supplementary Fig. 16). The distribution of copying proportions from donor clades along the chromosomes of isolates from clade VII (Fig. 6d) is consistent with the lack of recombination between donor haplotypes, suggesting that the origin of clade VII could be linked with somatic hybridization.

A somatic hybridization between divergent isolates was proposed previously as the most likely mechanism of the Ug99 isolate's origin, with Pgt21-0 carrying one of the donor haplotypes and the other donor remaining unknown[25]. By fitting a two-way admixture model ($r^2 = 0.72$), we identified that Ug99 race haplotypes from clade I share ancestry with isolates from clade II (87KEN3018-1, 84KEN8B, 09ETH8-3, and 99EGY4A) and clade IX (including Pgt21-0) (Fig. 6b, e and Supplementary Figs. 17 and 18). Since the previously established donor of the Ug99 haplotype, Pgt21-0[25], clustered with other isolates in clade IX (Fig. 5a), we hypothesized that isolates from clade II is the second likely donor source of the Ug99 haplotypes. The local ancestry along the Ug99 chromosomes shows an equal probability of copying from both donor groups (Fig. 6e and Supplementary Figs. 17 and 18), indicating that hybridization between these two sources contributed to the origin of Ug99. To validate this conclusion, we selected a set of 2320 homozygous SNP sites, which are fixed for alternative alleles in each of the two donor populations. Out of these SNPs, 2221 (95.7%) were heterozygous in Ug99 (Fig. 6f), which confirms the hybrid origin of Ug99 and suggests that an isolate closely related to the clade II contributed to the second Ug99 haplotype.

In summary, by examining the local ancestry of individual Pgt isolates from clades showing evidence of admixture, we showed that the origin of three clades (VI-A, UN1, and UN2) is linked with sexual recombination between the diverged Pgt lineages (Fig. 7 and Supplementary Figs. 12, 13 and 15). The other five admixed clades (I, II, VII, IX, and VI-B) displayed patterns of local ancestry consistent with somatic hybridization (Fig. 7 and Supplementary Figs. 16, 17, 19–20).

## Relationship between admixed origin and virulence in the Pgt panel

It has been hypothesized that both admixture and sexual recombination between isolates from distinct lineages may lead to increased levels of virulence[25,34,49,50]. To assess the impact of inter-lineage admixture on virulence, we compared the virulence scores (Supplementary Data 6) collected on a panel of wheat lines with 47 $Sr$ genes for the Pgt isolates from the source and admixed clades identified by MOSAIC (Fig. 7). For each trio of clades (one admixed and two source clades), we have identified resistance genes ($Sr$) that were effective against all Pgt isolates from both source clades. Within this group of $Sr$ genes, we counted how many $Sr$ genes became ineffective against at least half of the isolates from the corresponding admixed clade. This loss of $Sr$ gene efficiency could most likely be attributed to post-admixture processes affecting Pgt virulence. For example, out of 13 $Sr$ genes, which are effective against all Pgt isolates from source clades IX and II, four (30%) lost their efficiency against Pgt from the admixed Ug99 clade (clade I). Using this approach, we show that, on average, 15% of $Sr$ genes could have lost their efficiency after admixture among distinct Pgt lineages (Supplementary Table 15).

To assess the impact of sexual recombination on virulence, we compared the virulence profiles of the two isolates from races BFBJC and SCCLC and their $F_2$ progeny[51] on a set of six wheat lines carrying resistance genes $Sr7b$, $Sr11$, $Sr6$, $Sr9b$, $Sr30$, and $Sr24$, all effective against both parental races. Our analyses show that the virulence of some of the $F_2$ isolates on the panel of wheat lines increased compared to the parental isolates (Supplementary Table 16), which resulted in an overall increase in the virulence of the $F_2$ population (Fig. 8b). In a set of 20 recombinant $F_2$ isolates, 11 isolates showed virulence against $Sr6$, six isolates were $Sr30$ virulent, four isolates showed virulence against $Sr7b$ and $Sr9b$, two isolates acquired virulence against $Sr11$, and one $F_2$ isolate became $Sr24$-virulent. In summary, 18 out of 20 $F_2$ recombinants

became virulent to at least one of these six resistance genes. If we will assume that the loss of avirulence in the $F_2$ progeny is associated with the homozygosity for virulent alleles of genes, then the observed results could be explained by the presence of each of the avirulence loci in heterozygous state in both parental Pgt isolates and by the segregation of these loci in the progeny. In this case, we can expect higher levels of heterozygosity coinciding with the virulence of a smaller number of resistance genes.

We compared the level of heterozygosity and virulence to multiple resistance genes (Fig. 8c, d) between the North American and non-North American Pgt isolates (isolates from the $F_2$ population were excluded). The North American group of isolates showed lower levels of mean observed heterozygosity ($H_o = 0.16$), inbreeding coefficient ($F_{IS} = 0.03$), and genetic diversity ($\pi$) than isolates outside of the US ($H_o = 0.21$, $F_{IS} = 0.08$) (Fig. 8c). However, contrary to the expectation based on the analysis of $F_2$ population (Fig. 8b), the North American isolates were virulent on a smaller number of resistance genes than non-North American isolates (Fig. 8d). The positive relationship between the levels of heterozygosity and virulence suggests that the homozygosity for the virulent alleles of avirulence factors in the Pgt panel is unlikely the only factor driving an increase in virulence to stem rust resistance genes.

## Discussion

The comparison of the genomic distribution of the predicted effector-encoding genes and patterns of selection and genetic differentiation between the groups of Pgt isolates from distinct geographic regions suggests that effectors are more likely the selection targets of regional adaptation than other groups of genes. The signatures of selection around effector-encoding genes were evident for Pgt isolates grouped based on either virulence specificity or geographic regions, indicating that effectors could be associated with adaptation to new environmental conditions as well as adaptation to new genotypes of wheat cultivars. The latter is consistent with the earlier studies that showed that some fungal effectors display the signatures of selection in populations adapted to specific hosts[10,52–54].

A set of highly contiguous haplotype-resolved genome assemblies[25] generated using long-read sequence data allowed us to investigate the impact of SVs on effector repertoire in the Pgt diversity panel. The overlap between effectors and SVs that show the high levels of inter-population genetic differentiation is suggestive of SV's importance in adaptive evolution[55]. The overlap of effector-encoding genes and some of the known avirulence factors (AvrSr35[14] and AvrSr50[15]) with the regions affected by SVs supports this hypothesis.

Previously, the comparative genome analysis between different species of fungi showed that the distribution of effector-encoding genes across the genome is not random and tends to be physically associated with the TE-rich repetitive regions[21]. Though several studies of fungal genomes found no evidence for this "two-speed genome" architecture[25,56,57], our analyses of intraspecies genomic variation appear to be compatible with this model. We show that effectors tend to be enriched in the regions showing high levels of structural variation and TE content. This observation is consistent with the role of TEs in promoting structural variation through mechanisms associated with DNA recombination and replication processes[58,59]. Combined, our results indicate that the TE-driven SVs could have contributed to the evolution of the Pgt effector content.

The relative roles of sexual recombination and somatic hybridization in shaping Pgt diversity and the evolution of new virulence are poorly understood. Here, we model local fine-scale ancestry as a two-way admixture and show that chromosomal patterns of ancestry could be used to ascertain the contribution of sexual recombination and somatic hybridization to the origin of the Pgt lineages. We show that admixture between the distinct genetic groups of isolates is common in Pgt and that both sexual recombination and somatic hybridization

impacted the genetic composition of Pgt isolates. The MOSAIC analyses suggest that the origin of 8 out of 12 phylogenetic clades in our Pgt panel could be modeled as a two-way admixture. The analyses of local ancestry distribution along the chromosomes of individual isolates from each of these eight clades suggest that admixture events leading to the origin of clades I, II, VII, IX, and VI-B were likely associated with somatic hybridization (Fig. 7), whereas the origin of clades VI-A, UN1, and UN2 was likely associated with sexual recombination between the donor clades (Fig. 7).

Our results suggest that the origin of the highly virulent TKTTF race (clade IV) from Ethiopia appears not linked with a population admixture. However, considering the results of an earlier study, which suggested that clade IV isolates sampled in 2013 originated from the sexually reproducing population[33], it is likely that the TKTTF origin involves recombination between haplotypes within the same clade and step-wise accumulation of mutations in avirulence factors. The isolates from closely related clade VII, which share about half of its ancestry with clade IV, show evidence of two-way admixture, which, based on the chromosomal patterns of local ancestry, likely results from somatic hybridization between isolates from clades VI-A and VI-B (Fig. 7). Interestingly, the origin of clade VI-B is likely associated with somatic hybridization between clades VII and VI-C (Fig. 7). The reciprocal hybridization events inferred by MOSAIC between clades VII and VI-B suggest that these clades share the same haplotype, though the actual order of hybridization events could not be determined and needs further investigation.

Previously, it was shown that somatic hybridization also contributed to the origin of the Ug99 race, with one of the haplotypes tracing its origin to isolate Pgt21-0[25]. The origin of the second Ug99 haplotype remained uncertain. The analyses of local ancestry and clade-specific SNP alleles in our Pgt panel confirmed somatic hybridization as one of the most likely mechanisms of the Ug99 origin and Pgt21-0 as a possible donor of one of the haplotypes. This isolate belongs to a group of isolates from new clade IX, also including isolates from Ethiopia and Turkey. We identified the unknown source of the second Ug99 haplotype, which includes a group of isolates from Kenya and Ethiopia that are part of the previously defined clade II[45].

The analysis of ancestry segments in the isolates from clades VI-A, UN1, and UN2 revealed recombinant chromosomes, suggesting their origin in populations undergoing sexual reproduction. The role of sexual recombination in the origin of new virulence specificities was confirmed by phenotyping the progeny of two Pgt isolates[51] using the panel of wheat lines carrying the previously characterized stem rust resistance genes[60,61]. These results were consistent with earlier work suggesting that sexually reproducing Pgt isolates collected from barberry could be linked with rust epidemics[6] and high levels of virulence diversity[34].

There are several mechanisms by which sexual recombination and somatic hybridization could contribute to the origin of new virulent isolates. The loss of dominant avirulence factors in sexually recombining populations with some minor contribution from the suppressors of avirulence genes were previously postulated as the main driving factors behind the origin of virulent Pgt isolates[51]. One of the expected consequences of recent selection for recessive alleles of avirulence genes in populations with the prevalence of asexual reproduction is genome-wide loss of heterozygosity and diversity. These diversity trends were observed for the US sample of Pgt isolates and are consistent with the prevalence of asexual Pgt reproduction in the United States[39] due to the eradication of the alternate Pgt host, barberry. Using these mechanisms, however, it is harder to explain the loss of efficiency against admixed Pgt isolates observed for 15% of Sr genes, which otherwise are highly effective against ancestral source Pgt clades. Though, it is possible that admixture could promote virulence by bringing together two haplotypes carrying virulent alleles of effectors unrecognizable by the host's immune system. Alternatively,

we could consider the scenario where broader virulence of admixed isolates is associated with their improved ability to suppress the host's immune responses. In this case, admixture likely promotes virulence by combining avirulence suppressors evolved in distinct lineages. This strategy of new virulence origin was proposed for inter-specific hybrids of *Blumeria graminis* and *Zymoseptoria tritici*[62,63]. To what extent it is applicable to virulence evolution at the intra-specific level remains to be investigated. Further studies aimed at cloning and characterization of pathogen effectors from diverse pathosystems are needed to uncover the entire diversity of mechanisms that contribute to the evolution of virulence in agricultural pathogens. Resources generated in our study, including sequenced and phenotyped Pgt isolates, have the potential to facilitate the characterization of the Pgt avirulence factors, deepen our understanding of how this important pathogen evolves and adapts, and ultimately contribute to efforts aimed at reducing the risk of Pgt pandemics.

## Methods

### DNA extraction and sequencing for 99KS76A-1

The wheat cultivar "Morocco" was used for spores grow-up and collection of 99KS76A. To obtain both high quality and high quantities of fungal DNA, collected urediniospores were dusted in a fine layer over the top of the diluted germination solution using a 40 μm metal sieve, and germinated spore mats were collected after 16 h. A high-quality fungal DNA isolation protocol[64] was used, with minor modifications. We used a lysis buffer prepared by mixing 5 ml of Buffer A, 5 ml of Buffer B, 2 ml of Buffer C, 1 ml of 10% PVP 40, and 1 ml of sterile ddH$_2$O. Buffer A composition: 0.35 M sorbitol, 0.1 M Tris-HCl (pH 8.0), 5 mM EDTA (pH 8.0) (autoclaved to sterilize). Buffer B composition: 0.2 M Tris-HCl (pH 8.0), 50 mM EDTA (pH 8.0), 2 M NaCl, 2% CTAB (autoclaved to sterilize). Composition of Buffer C: 5% Sarkosyl *N*-lauroylsarcosine sodium salt (Sigma L5125) (filtered through 0.22-micron membrane to sterilize). The lysis buffer was heated to 50 degrees after mixing and cooled to room temperature. Samples were grinded and added to the lysis buffer before adding 50 μl of RNAse A (10 mg/ml). Samples were rotated on Rotator Genie at the slowest speed at room temperature for 1 h, followed by adding 200 μl of Proteinase K and rotating for an additional 1–2 h. The tube was incubated for 5 min on ice, and then 200 μl of Proteinase K was added, mixed, and incubated for another 5 min. The samples were centrifuged at 5000×g for 12 min at 4 °C, supernatant was removed and placed into a new tube after adding 15 ml of phenol/chloroform/isoamyl alcohol (P/C/I, 25:24:1), followed by rotating for 5 min at the slowest speed. The sample was centrifuged at 4000×g for 10 min at 4 °C, followed by removing the supernatant and adding 15 ml P/C/I. Sodium acetate and isopropanol were added and mixed, followed by incubation at room temperature for 5–10 min, and centrifugation at 8000×g for 30 min at 4 °C. Supernatant was poured slowly, and the pellet was washed with 70% ethanol and dissolved in 100 ul of 10 mM Tris-HCl buffer (pH 8.5)[64].

Oxford Nanopore (OXN) LSK-109 ligation sequencing kits with flow cell 9.4.1 was used to generate sequencing data for 99KS76A-1. Long-read DNA sequence data were generated using five flow cells using a MinION device. Raw fast5 files were converted to fastq files using guppy software. Resulting fastq reads were passed through the NanoFilt program to retain sequencing reads with a quality score >6 and read length >1500 bp.

For Hi-C sequencing, a spore mat was sent to Dovetail Genomics for library preparation and sequencing. In addition, previously published Miseq and PacBio data developed for the 99KS76A-1 isolate[14] were used in this study.

### Genome assembly of the 99KS76A-1 isolate

To create haplotype-resolved assembly of the 99KS76A-1 isolate (race RKRQC), we have used the combination of High-throughput Chromosome Conformation Capture (Hi-C) technique, long-read Oxford Nanopore (OXN) sequencing with short-read Illumina sequencing (2 × 300 bp). The previously generated Illumina MiSeq reads (2 × 300 bp)[14] were trimmed using Trimmomatic v0.38[65] with the following parameters: LEADING:3 TRAILING:3 SLIDINGWINDOW:4:15 MINLEN:36. A total of 163,655,621 Hi-C read pairs, 4,175,267 OXN reads with the average length of 4258 bp, 264,532 PacBio reads with an average length of 2350 bp, and 31,390,809 Illumina read pairs (2 × 300 bp) were generated. Assembly was performed using Canu v1.7[26] followed by correcting assembly errors by aligning MiSeq reads (2 × 300 bp) to the assembled contigs using Pilon v1.22[19] (4.36% error rate of the assembly was corrected). For this purpose, the trimmed MiSeq reads were aligned to the Canu assembly using BWA-mem v0.7.17[66] with default settings and sorted with the samtools v1.8[67].

A two-step pipeline was utilized for scaffolding the contigs generated by Canu. The first step was based on a bin-assignment strategy[25]. The gene models predicted in our 99KS76A-1 assembly were aligned against the assembled contigs using BLASTN v2.9.0[68] and results were filtered to retain alignments with identity ≥95% and gene coverage ≥50%. In *Puccinia* spp., somatic cells are dikaryotic, consisting of two physically distinct haploid nuclei[25,56,57]. If two contigs shared at least 25 genes, they were considered to represent haplotypes from distinct Pgt nuclei. These homologous contigs were grouped into distinct bins (Supplementary Table 1). In total, we identified 33 bins, including 120 contigs, with the total contig length of ~150 Mb. Sequence collinearity between contigs in each bin was assessed using MUMMer v4.0[36] with nucmer (--maxmatch) and plotted by mummer-plot (Supplementary Fig. 23). Step two was based on the ALLHiC assembler v0.9.8[27,28] that started with raw Hi-C reads mapped to the Canu contigs using BWA-MEM v0.7.17[66]. The contigs representing allelic variants from distinct nuclei were identified by aligning gene models from the Canu assembly to the assembled chromosomes of the Pgt21-0 isolate[25]. This information was also used to group contigs into 18 homologous groups (HGs). For each HG, noisy inter-allelic and weak Hi-C density signals (number of read pairs/length (contigA + contigB) <0.35) were removed by the ALLHiC Pruning function. Subsequently, together with information gathered from the previous bin-assignment step, contigs in each HG were partitioned into two groups (or haplotypes) through the ALLHiC Partition routine. The ALLHiC[27] scaffolding resulted in a chromosome-level assembly of the 99KS76A-1 genome consisting of 36 chromosomes with a total length of ~161 Mbp (Fig. 1a). The assembled chromosomes were assigned to haplotypes from distinct nuclei using Hi-C data. This approach has been previously applied for assembling the haplotype-resolved assemblies of *Puccinia triticina* and *P. graminis*[25,29]. Out of 36 chromosomes, 34 could be assigned to distinct haplotypes. The 99KS76A-1 haplotypes were designated as "E" and "F" (Supplementary Fig. 1), which showed 97.1 and 96.8% similarity, respectively, to haplotype A of Ug99, and 96.7 and 97.4% similarity, respectively, to haplotype C of Ug99 (Fig. 1b).

The haplotype assignment for the 99KS76A-1 genome scaffolds was performed by counting Hi-C links between the scaffolds (Supplementary Fig. 1). Hi-C reads were processed with HiCUP pipeline v0.8.2[69] to filter out non-unique alignments and experimental artefacts, using Bowtie2[70] as aligner and the 99KS76A-1 assembly as a reference. The resulting haplotypes correspond to genomes of individual nuclei in a dikaryotic fungus. The assembled genome KSU_Pgt_99KS76A_2.0 is available at NCBI (accession number PGFY02000000).

### Assessment of the 99KS76A-1 genome assembly quality

Completeness of the 99KS76A-1 assembly was assessed by BUSCO[30] in the genome mode using a basidiomycota_odb10 set of genes. The Pgt21-0 assembly was aligned against the chromosome-level assembly of 99KS76A-1 using MUMMer v4.0[36] with nucmer (--maxgap = 10,000), and alignment results were plotted using Dotplotly (-l -m 10000) to evaluate the collinearity between assemblies (Supplementary Fig. 2). We also aligned the 99KS76A-1 assembly against itself using the

MUMmer's nucmer with settings --maxmatch --maxgap = 10,000 to validate matches between the haplotypes (Supplementary Fig. 3).

## Gene prediction and functional annotation

RepeatMasker v4.1.0 with Repbase v. 25.12. was used to mask transposable elements in the final assembly. The evidence-based gene prediction was performed with the PASA pipeline v.r20140417[71]. For gene prediction we used (1) 295,186 transcripts previously assembled using the RNA-seq data generated for the Pgt-infected leaf tissues (includes both the wheat and Pgt transcripts)[14], (2) 37,843 CDS reported for Pgt21-0 genome[25]; (3) 37,820 CDS reported for ug99 genome[25]; (4) 15,979 gene models reported for the genome of Pgt strain CDL 75-36-700-3[32]; (5) 22,391 transcripts that were generated for the Australian Pgt isolates[31]. Genes encoding effector proteins were predicted using SignalP v5.0[72], ApoplastP v1.0.1[73], and Phobious v1.01[74], with the removal of proteins targeted to mitochondria or having transmembrane domains by running TargetP v2.0[75] and TMHMM v2.0[76,77].

## Variant calling in the Pgt panel

A diverse collection of 77 rust isolates (Supplementary Data 2) from the USDA Cereal Disease Lab (CDL) was sequenced using paired-end sequencing run on HiSeq2500 (2 × 150 bp). DNA was extracted as previously described[33]. Illumina reads were aligned to the non-redundant set of the 99KS76A-1 scaffolds using HISAT2 program[78] with option --no-spliced-alignment. After removing reads mapping to multiple locations and marking duplicates, multi-sample variant calling was performed using the HaplotypeCaller of GATK v4.1.4.1[79] with the default settings. The variable sites have been filtered to remove those that (1) had more than two alleles, (2) minimum depth of total read coverage was below 3, (3) more than 25% of isolates have no genotype calls, or (4) minimum depth of alternative allele read coverage below 3. To improve the accuracy of rare variant calling with the minor allele present in only single isolate (singleton variants), we removed sites with a minimum depth of read coverage below 10. To assess the accuracy of genotype calling, we compared the allelic variants called for 99KS76A-1 by mapping short-read data to the reference genome of 99KS76A-1. Both SNP and indel calling showed 99% concordance, indicating the high accuracy of variant calling in our dataset.

## Genetic diversity analyses

The Pgt panel includes 18 $F_2$ isolates derived from a biparental cross between isolates CRL 78-21-BB463 and 75-36-700-3[51]. These isolates were excluded from most of the genetic diversity analyses. Minor allele frequency (MAF) for each SNP site was computed using BCFtools v.1.9 fill-tags function. R package PopGenome v2.7.5[80] was used to calculate nucleotide diversity ($\pi$), TajimaD, and pairwise $F_{ST}$ statistics. In these calculations, sites with missing data or located within the regions overlapping with repeated sequences identified by RepeatMasker v4.1.0 were excluded.

LD analyses were performed on a dataset after removing variants with MAF <0.025 and heterozygous SNPs. Plink v1.9[81] was used to randomly pick 5% of SNPs for LD calculation with the following parameters: --ld-window 20000, --ld-window-kb 7000, --ld-window-r2 0, --allow-extra-chr, --keep-allele-order, --thin 0.05. To reduce the effect of population structure on the estimates of the rate of LD decay, $r^2$ was calculated for two samples of Pgt isolates (15 from the United States and 22 outside of the United States) that show no evidence of admixture at the optimal value of $K = 6$. LD decay was modeled by fitting the estimates of $r^2$ to a model described in ref. 38.

## Selection scans

To reduce the rate of detecting false selection events, we conducted selection scans using two different approaches implemented in XP-CLR[40] and SweeD[41]. Both methods were shown to be robust to variation

in demographic history and recombination rate heterogeneity across the genome. The datasets used for both XP-CLR and SweeD analyses were filtered by vcftools with −remove-indels −maf 0.01 and −max-missing 0.75. XP-CLR[40] was run with a grid size of 100 bp, sliding window size of 0.05 cM, maximum of 50 SNPs within each window, and pairwise correlation coefficient of 0.8. The SweeD was run with the same grid size of 100. To select test statistic thresholds for detecting outliers, we conducted selection scans using both approaches in the simulated datasets, taking into account the demographic history of the two Pgt samples, 15 from the United States (US isolates) and 22 from Africa, Europe, and South America (non-US isolates). The two sets of Pgt isolates were selected to be genetically uniform and have no admixed genotypes at the optimal value of $K = 6$. The population size and split time history of these Pgt samples was inferred using the SMC + + tool[82] using the process described in the software manual. The effective population sizes of non-US and US isolates were estimated to be 124,000 ($N_0$) and 50,840 ($N_1$), respectively, with a split time of 95,000–110,000 generations ago. These estimates of population history were used to simulate diversity datasets for 37 individuals (22 individuals in population 1 and 15 individuals in population 2) using the following MaCS[83] command: macs 37 5e5 -i 1 -t 0.003 -r 0.00001 -I 2 22 15 -n 2 0.41 -ej 0.22 2 1 2. Effective population size-scaled diversity −t was calculated using the mutation rate of $5 \times 10^{-9}$[84]. The 4N-scaled recombination rate parameter -r was tested for ranges between 0.001 and 0.00001 and shown not to have a significant impact on the results of selective sweep statistics. Thus, we used a more conservative approach to simulate data with the lower value of -r 0.00001. The effective size of population 2 ($N_1$) was 0.41 of the effective size of population 1 ($N_0$). For each simulated dataset, we calculated Tajima's D and diversity statistics using msstats program based on the C + + libsequence library[85] and selected for downstream selective sweep analyses only those sets that match the observed diversity statistics within one standard deviation of Tajima's D (0.74 ± 0.75) and diversity ($3.3 \times 10^{-3}$ ± $3.0 \times 10^{-3}$). The selective sweep analyses using both methods were conducted for a total of 1000 datasets. The 95th percentiles of the distributions built using the highest test statistic values from each simulated dataset were 53.48 for XP-CLR and 4.27 for SweeD. Since these values were lower than the 95th percentiles of the test statistic distribution for XP-CLR (72.26) and SweeD (17.68) in the real-life dataset, the empirically defined thresholds were used in the study. The outliers above the XP-CLR and SweeD scan thresholds were identified, and the adjacent outlier regions were merged. The regions shared by both scans were classified as genomic regions under selection and the number of effector-encoding genes overlapping with these regions was calculated. To assess the chance of obtaining the set of genomic regions overlapping (159) with effector-encoding genes, we counted the number of genomic regions overlapping with effectors in 1000 randomized datasets (Fig. 3c).

## Structural variant detection and diversity analyses

To detect structural variants among the four haplotypes of the 99KS76A-1 and Pgt21-0 genomes (two haplotypes per genome), four homologous chromosomes were compared to each other with MUMmer v4.0[36] nucmer (--mum). To improve structural variant detection, two methods were used. First method is based on Assemblytics[42]. After filtering the MUMmer results with parameters setting of -i 90 -r -q, Assemblytics[42] was used for detecting SVs with unique anchor lengths of 500 bp, minimum variant size of 50 bp, and maximum variant size of 1000 kb. Since Assemblytics is more accurate in detecting small-size SVs than the large ones, the large SVs with size ≥30 kb were filtered out if they did not appear as consecutive in terms of coordinates along the chromosome. The large-size SVs (≥10 kbp) were identified using the second method. In the filtered MUMmer alignment results, only those alignments that showed consecutive coordinates were kept, and gaps between the alignments were

considered SVs if their size was between 10 and 1500 kb. SVs obtained using these two methods were combined for further analysis. There are four types of SVs were detected using both methods: insertions, deletions, expansions, and contractions[42] (Supplementary Fig. 5).

We assessed the accuracy of SV calling by comparing regions carrying SVs with individual long OXN reads spanning these regions in the 99KS76A-1 genome. Since each OXN read represents a single DNA molecule, this approach provides independent validation of identified SVs. For this analysis, we selected 1947 heterozygous SVs from ten homologous chromosomes (1E-5E, 1F-5F) of 99KS76A-1. The OXN reads were mapped to both haplotypes of the 99KS76A-1 genome using minimap2, and those reads that span an entire SV and at least 1000 bp from each side were compared with the SV alleles detected by the alignment of the Pgt haplotypes. Out of 1889 analyzed SVs, 95% were confirmed to have both SV alleles at each variable site, indicating that the SV discovery by whole genome comparison has good accuracy. The actual concordance rate between the SVs detected by haplotype and OXN read alignments is likely higher. Due to the repetitive nature of some of the OXN reads and the high per-read error rate, some SVs were covered by reads mapping to more than single locations. These SVs were counted towards unconfirmed SVs.

To genotype SVs in each Pgt isolate, sets of consecutive diagnostic 31 nucleotide-long k-mers that cover the boundaries SV sites were developed (Supplementary Fig. 5). Because SNP variation at the SV boundaries in the Pgt panel could affect the results of k-mer analysis, we have designed k-mers only to those SVs that has no SNPs around their boundaries, resulting in a set of 10,188 SVs. With the k-mer database built for each isolate in the Pgt panel using Jellyfish v2.2.10[86], the presence or absence of these diagnostic k-mers was recorded to generate a matrix of SV genotypes for the whole Pgt panel using the following rules: (1) if the number of k-mers for the SV presence site (which include two boundaries) ≥15, and the number of k-mers for the SV absence site (which includes only one boundary) = 0, then genotype is 1/1; (2) if the number of k-mers for the SV presence site ≥15, and the number of k-mers for the SV absence site ≥5, then genotype is 0/1;3) if the number of k-mers for the SV presence site <5, and the number of k-mers for the SV absence site ≥5, then genotype is 0/0. The genotyping matrix was converted to the vcf format for constructing the NJ tree after filtering out SVs with no missing rate and minor allele frequency ≤10%.

The accuracy of k-mer-based SV genotype calling in the short-read Illumina datasets was assessed for SVs detected in the 99KS76A-1 genome. We compared the SV calls generated using the k-mer approach with SV calls identified by the haplotype and OXN-read alignment and showed these two datasets are 99.9% concordant.

We calculated the correlation between the proportion of the Pgt genome affected by SV, and the TE and effector-gene content. For this purpose, the size of the genome affected by SV and TEs, and the number of effector-encoding genes was calculated within the non-overlapping windows of 1 Mb spanning the entire genome. For comparing the MAF spectrum of SVs and synonymous SNPs detected between 4 haplotypes of 99KS76A-1 and Pgt21-0, SV types corresponding to insertion and deletion (because their genotype were more accurate than expansions and contractions) were filtered to remove sites with more than 10% missing rate. The SNP dataset comparable to SVs was built by using only those sites that segregated between the four haplotypes of 99KS76A-1 and Pgt21-0. These sites were detected by comparing the 99KS76A-1 and Pgt21-0 haplotypes using MUMmer[36] with the following settings: show-snps -Clr -I -T. The genotypes at these SNP sites were extracted from Pgt panel. vcftools was used to calculate the pairwise $F_{ST}$ between geographical regions for SVs and SNPs within the window size of 1 Mbp.

## Population structure
In addition to isolates in the Pgt diversity panel, we have also included CDL 775-36-700-3 and Pgt21-0 for population structure analysis. For

these two isolates, the genotypes of SNP sites segregating in the Pgt diversity panel were inferred by aligning 100 bp SNP flanking sequences to the published genome assembly[32]. SNP positions were identified using the blastn2snp tool from JVarkit v.1.0. Since the CDL 775-36-700-3 genome assembly[32] is largely haploid with the contigs from both nuclei collapsed together, genotypes were recorded as either 1/1 or 0/0 when only one of the two possible alleles is present at the aligned site. In case if no aligned sequences were detected or aligned sequences represented both alleles, genotypes are called missing.

To construct a neighbor-joining (NJ) tree, SNP sites are filtered to keep those that have no missing data, and a minor allele present in at least three isolates. Distance-based phylogenetic analysis was computed using R v4.0.0, with vcfR v1.12.0[87], adegenet v2.1.3[88], ape v5.4[89], and ggtree v2.2.4[90] packages. Euclidean genetic distance between each individual isolate was calculated with the function "dist" in adegenet. NJ estimation of Saitou and Nei[91] was built with "nj" function of ape, and was plotted using ggtree[90].

ADMIXTURE v1.3.0[44] was performed with fivefold cross-validation (CV) for K ranging from 2 to 15 to estimate the optimal number of populations. At each value of K, ADMIXTURE was run ten times with different starting random seed numbers. Q matrix for the same values of K was merged using mean values and plotted using the R package pophelper v2.3.0. For identifying potential donor sources for each clade in the whole panel, SNPs were filtered by vcftools with --maf 0.1 and −max-missing 1, and phased by Beagle.28Jun21.220.jar[92] with default settings. For inferring the local ancestry of each clade using donor sources, SNPs were phased and filtered by vcftools with --maf 0.05 and −max-missing 1 for each dataset. MOSAIC was used to estimate the copying probability of donor sources and infer local ancestry along chromosomes with default settings (Supplementary Data 5). The geographic inbreeding coefficient was calculated with vcftools (v0.1.13) -het[93], and $\pi$ in different geographic regions was calculated with PopGenome[80] based on gene models.

## Phenotyping Pgt isolates on the panel of differential wheat lines
The virulence of each of the Pgt isolates was assessed by infecting the panel of wheat differential lines carrying known stem rust resistance genes (Sr). The wheat seedlings were infected and scored at the USDA Cereal Disease Laboratory (CDL) at a biosafety level-3 facility (University of Minnesota). The Pgt urediniospores stored at −80 °C were heat-shocked for 6–10 min in a water bath at 42 °C and resuspended in Soltrol 170 light oil (Chevron Phillips Chemical Company, The Woodlands, TX). Inoculations were performed by air spray at the one-leaf stage. Immediately after inoculation, plants were transferred to a dew chamber and incubated in the dark for 16 h at 22 °C and under 100% relative humidity. The seedlings were transferred to the greenhouse and grown at 22 °C day and 18 °C night with 16 h of photoperiod. The Pgt virulence scores ranging from 0 (highly avirulent) to 9 (highly virulent) on each differential were assessed 12–14 days after inoculation. According to the North American Pgt nomenclature[60], the infection types induced by Pgt isolates were considered H (High) if the score was less than or equal 6 and L (Low) if the score was higher than 6.

## Reporting summary
Further information on research design is available in the Nature Research Reporting Summary linked to this article.

## Data availability
The full genome assembly of Pgt isolates 99KS76A-1 is available at NCBI under accession number PGFY00000000.2 and GCA_002762355.2 associated with BioProject PRJNA313186. Sequencing data used for genome assembly, including Oxford Nanopore SRR18170993, PacBio SRR18170992, Hi-C SRR18170990, and Miseq SRR18170991] reads, have been deposited to NCBI under

BioProject PRJNA313186. The Hi-C contact frequency data is available from NCBI GEO GSE210552. RNA-seq or CDS data for genome annotation were downloaded from NCBI BioProjects PRJNA415853, PRJNA516922, PRJNA66375, and PRJNA347320. Illumina sequence files for the Pgt diversity panel were deposited to NCBI under BioProject PRJNA803546. Variants calls are available from the EBI EVA under accession PRJEB56443. Phenotypic data were provided in Supplementary Data 6.

## Code availability

A custom script used in the study is deposited to GitHub: https://github.com/akhunovlab/stem_rust_diversity[94].

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

## Acknowledgements
This project was supported by a grant from the Bill and Melinda Gates Foundation (INV-004430).

## Author contributions
Y.G. assembled and annotated the genome, analyzed diversity and phenotypic data, and helped with drafting the manuscript; B.B. prepared Pgt samples and DNA for Hi-C and whole-genome sequencing; A.S. generation of Pgt samples for genome sequencing; F.H. contributed to initial processing and analysis of NGS data; R.L.B. contributed a collection of Pgt isolates, data interpretation, and manuscript revision; J.P.F. preparation of Pgt samples for Hi-C and genome sequencing contributed to developing a genome assembly strategy; A.A. designed and conducted NGS experiments; K.W.J. OXN library construction and data generation; M.N.R. disease resistance scoring for the Pgt panel; L.J.S. contributed Pgt isolates, generated NGS data for the diversity panel, interpreted results, and helped to draft the manuscript; E.A. conceived idea, interpreted results, coordinated project, and wrote the manuscript. All authors read the manuscript and approved the final version.

## Competing interests
The authors declare no competing interests.

## Additional information

**Correspondence and requests** for materials should be addressed to Les J. Szabo or Eduard Akhunov.

