## [Peer Review File · Nature Communications]

Population genomics of *Puccinia graminis* f.sp. *tritici* highlights the role of admixture in the origin of virulent wheat rust racesReviewers' Comments:

Reviewer #1:

Remarks to the Author:

The manuscript by Guo et al. describes genome analyses of the wheat stem rust *Puccinia graminis* f. sp. tritici. This pathogen is a major concern to wheat production worldwide and is often used to illustrate the breakdown of effective resistance in crops (i.e. the emergence of the Ug99 lineage among stem rust populations). Genomic analyses of the species were initiated with a draft assembly by Duplessis et al (2011) and high-quality haplotype-resolved assemblies by Li et al (2019, Nat Comm). The dikaryotic lifestyle (two possibly distinct nuclei present per cell) and the repetitive genome caused challenges prior to the latest round of genome sequencing innovations. This study here adds a third high-quality genome assembly for the species and documents structural variation within the species affecting putative effector complements. The authors then focus on identified structural variant loci to genotype a broader panel of resequenced genomes. The study also expands on the work by Li et al. by finding more complete evidence for global admixture, possibly links with virulence, somatic hybridization and recombination.

The study tackles many different aspects of the genetic make-up of the species and explores links with virulence, as well as identifying putative admixture events. The results text could be significantly shortened so that the focus is on key independent findings. The current version is lengthy and has sections that are largely redundant or could be described more succinctly to amplify the message.

Main comments :

- The haplotype resolution assembly strategy for the US isolate follows a well-tested pipeline following Li et al in 2019 that produced very well resolved haploid genome assemblies. The presented data shows that the new haplotype-resolved assembly is also of high quality. A minor point would be to report the BUSCO findings per haplotype in Fig 1c. My understanding is that the high number of duplicated BUSCO genes comes from having both haplotypes combined for the analyses, correct?
- The results text in the first paragraphs would flow better though if only the essence would be kept and detailed numbers, mentions of software (incl. versions), read length, etc. only mentioned in the methods section.
- Reporting BUSCO scores for annotated genes would be helpful to assess to the completeness of the annotation.
- L197 – a more explicit statement on the global sampling directly in the results text would be helpful.
- L218-223: I agree that the bias towards rare variants for high-impact mutations is caused most likely by purifying selection. Allele frequency spectra can be strongly affected by the demographic structure though, hence dividing the isolates into reasonably homogeneous genetic pools prior to spectra analyses would be the appropriate approach to draw broader conclusions.
- Linkage disequilibrium decay curves should be fitted using e.g. the model by Ingvarsson (Genetics 2015) to properly account for population genetic processes influencing r^2 . The figure panel 2d seems to show a simple smoothing function.
- The population genetic analyses based on π , Tajima's D and F_{st} show highly significant differences but only minute effect sizes. This important caveat is not made clearly enough in the results text.
- The property of the allele frequency spectrum expressed by Tajima's D is strongly influenced by the demographic history of a population (expansion/contraction) and subdivisions. Only in ideal and at-

equilibrium populations can Tajima's D be used to identify positive versus balancing selection signatures. Looking at later analyses (i.e. Fig 5) shows that the grouping by regions produces highly heterogeneous populations unsuitable for Tajima's D analyses. Please address also concerns about using Tajima's D elsewhere in the manuscript.

- The Tajima's D and Fst analyses would be more coherently motivated if the authors first performed unsupervised clustering (as in Fig. 5) and use the information on genetic identities to create genetically homogeneous populations for selection scans / Tajima's D analyses.

- The XP-CLR and XP-EHH analyses are also sensitive to demographic properties. To retain such analyses in the manuscript, demographic models need to be implemented to provide a null expectation for selection signals expected from demographic factors alone. Signals in excess of such differentiation should then be used as the most likely selection signals instead of a 95th-percentile threshold. Depending on the demographic history either much less or much more loci might actually show signatures of selection. Here, carefully selecting homogeneous genetic groups is important instead of relying purely on geographic regions.

- L296: dN/dS is not a ratio of non-syn. vs syn. mutations but rather a difference in rates in a phylogenetic context. The authors may mean pN/pS instead? Using dN/dS within species is controversial due to confounding effects of within-locus recombination.

- L305: "Local adaptation" is not typically defined as high genetic differentiation of populations but selection of locally fit genotypes in the face of gene flow among populations thus generating GxE interactions. Please clarify the wording here depending on the actual aim of your analyses. My understanding is that you are looking for selection signals specific to individual genetic groups. Please see above for issues with using XP-CLR and the need for demographic modelling.

- L332: Does "more successful" refer to simple higher recovered numbers or challenges in recovering small vs. long SVs?

- How were the SVs retrieved from MUMmer and Assemblytics then quality filtered? Were any efforts made to identify rates of false positives or negatives by comparing with independent approaches? Please provide some insight into the SV quality assessment and add a quality test.

- The authors construct a large panel of SVs segregating among the fully assembled genomes. To overcome challenges of SV calling with short reads, k-mers were identified that should tag known SV alleles. I do not find a rigorous assessment how well this genotyping method performs. What is the likelihood that the chosen k-mers are mutated in the panel of Illumina sequenced strains? Mutated k-mers could lead to false negatives either at a certain rate. Similarly, what is the likelihood for false positive calls given the complexity of the genome and the amount of short read data processed? Assessing only the uniqueness of the k-mers in the two reference genomes seems insufficient here. Both false positive and false negative rates should be assessed given the novelty of the approach. Alternatively, I would suggest to apply and validate well tested graph-based methods (e.g. PanGenie, Giraffe, etc).

- L365: Please reword "accurately" as there seems to be no formal test associated with this statement.

- L381-382: Candidate loci for divergent selection among populations are typically identified using an outlier approach. Here, the authors seem to simply compare SNP- vs SV-based Fst values. This suggests that SV tend to be overall more differentiated among populations but not that SVs as a whole should be adaptive as written.

- Figure 5c: I find it impossible to relate the several dozen resistance gene labels in the legend text to the appropriate column in the figure itself. Please consider splitting this information into multiple

figures (or SI). Furthermore, what should the reader take from this matrix as a message?

- L496: To detect admixture between populations using local ancestry, the authors based themselves on phylogenetic clades defined in previous publications. My reading of Figure 5a is that the match between these phylogenetic clades and genetic cluster assignments based on ADMIXTURE is not perfect. Therefore, I am not following why phylogenetic clades based on limited markers should better reflect genetic structure than whole-genome based ADMIXTURE analysis. My concern here is that using these weakly defined phylogenetic groups for admixture analyses can lead to false positive and false negative admixture signals. Relying on genetic clusters instead should largely solve this issue.

- Figure 7 and virulence assays. What are the fine dots in panel a)? My understanding is that there are only a few dozen admixed and non-admixed individuals. Please report the statistics in full. I am unsure to understand how such a low p-value was found given a fairly small sample size. The same applies to panels c) and d) where showing individual dots would make the analyses more transparent.

- Code availability: Please provide custom scripts mentioned in the text on a public repository.

- Data availability: Where was the RNA-seq data used for the annotation deposited?

Reviewer #2:

Remarks to the Author:

Guo et al. present the assembly of a haplotype resolved *Puccinia graminis* f. sp. *tritici* genome by using several sequencing data types. This new resource (a great achievement in itself) was then leveraged to identify and characterize structural variation across a Pgt panel, as well the impact of structural variations on effector gene loci. It was determined that effectors are enriched in TE-rich regions that also have a high rate of structural variation, which may lead to rapid change/modification of effector loci and aid the pathogen in evading R-gene detection. A unique aspect of this manuscript that shed new light on the emergence of novel, virulent races was the examination of local ancestry. Importantly, the somatic hybridization event that led to the generation of the Ug99 lineage was confirmed and a potential second donor lineage was identified. Overall, this manuscript was very well written and the methods/analyses used were appropriate. This work advances our understanding on how complex genome dynamics may impact host-pathogen interactions, as well as details the manner in which somatic hybridization and sexual recombination contribute to the formation of new pathogen lineages. I have outlined some suggestions and questions for further clarification for the authors below.

L29: change to "We utilize long-read sequencing"

Supplementary Table 1: What is the connection to Supplementary Figure 12?

L125: Prior to this section, it may be helpful to briefly mention basic assembly statistics from the primary Canu assembly (or reference Supplementary Table 2 in some manner).

L178-181: Are the single-copy genes located randomly throughout the genome or are there large stretches of low haplotype divergence?

L214-215: What differentiates intergenic and upstream/downstream polymorphisms? Are upstream/downstream within a specific proximity of a gene?

Figure 2a: Why is exon and intergenic illustrated twice on this figure panel?

L221: I think this conclusion comparing nonsynonymous and synonymous SNPs may be a bit of a stretch, since the MAF distributions are very similar and not significantly different. I suggest shifting the focus of this section to the comparisons to the frequency spectrum of the higher impact mutations.

L218-223: Although both stop codon and splice site mutations are considered high impact mutations, I feel it may be more relevant to compare these categories separately.

L240-243: As the authors alluded to here, the existence of population structure can influence the calculation of LD. I suggest calculating LD decay for each population separately (for those with

sufficient representation), in addition to the total population presented here, to obtain a better estimate.

L248: change to "Tajima's D"

L312: Are these 14,436 outliers windows or combined adjacent genomic regions?

L314: The number of effector genes overlapping with randomly sampled genomic regions was still unclear to me. I suggest explicitly stating this number to clarify the comparison to the 818 effector genes that overlap outlier regions.

L319: To what degree were the results from the XP-CLR and XP-EHH tests correlated? Did some of the same conclusions drawn from the XP-CLR test hold true for the XP-EHH test (particularly the enrichment of effectors)? Do these tests differ in their abilities to detect hard vs soft sweeps?

L305-327: Although it was nice to highlight previously validated effectors identified in these selection analyses, I think it would be informative to elaborate further on other (presently unvalidated) effector loci that these tests detected. Were there any candidate effectors that showed even stronger evidence for positive selection than AvrSr35 or AvrSr50?

L383: Here it is stated that 560 SVs overlapped with 707 predicted effector genes, but in L338, it is stated that 806 effectors were affected by SVs. Is the former estimate from the Pgt panel and k-mer analysis, and the latter from just the comparison of 99KS76A-1 and Pgt21-0? Please clarify.

L396-400: Is the prevalence of Sr35 and Sr50 resistance genes in local cultivars known for these regions? If so, it would be interesting to discuss this in regards to these results.

L450: change to "individuals"

L819: How was de novo transcript assembly conducted?

Reviewer #3:

Remarks to the Author:

In general, I enjoyed reading this manuscript. It is well written, the conclusions are supported by the analyses, and most results are clear from the text and figures. I believe that the community will find these results of interest, as they build from previous research and add more insights into the genomic evolution of an important pathogen. I nevertheless have some comments mostly related to clarity improvement that should be addressed in revisions.

The introduction contains the needed information to understand the questions, context, and results overall. However, the structure of the text makes it hard to really understand it, the paragraphs alternate between information that is very specific to the species studied (ex.: paragraphs 1 and 3) and generalities about fungal pathogen genomes (ex.: paragraphs 2 and 4). Reorganizing the paragraphs to improve the flow would help the reader.

In several cases, method information is presented outside of the methods section in such detail that it hinders the general understanding of the workflow used. This is already the case in the last paragraph of the introduction: it is not necessary at this point in the manuscript to know that SV was detected specifically with kmers of 31 bp, for example. The first section of the results contains text that is almost word for word the content of the corresponding material and methods (from l.120 to 145). At this point in the text, the reader needs more general information and less technical details. As an example, what are Hi-C reads and how are they helpful in assembling genomes is more relevant than the specific version of the software used to analyze them, which is clearly at its place in the methods. Some points of the process are not very clear either: line 134/135 it seems that the Pgt21-0 assembly was used to gather contigs into groups. How meaningful then is the collinearity observed in line 146 between the two assemblies if the one was indeed used to build the other? Which BUSCO mode was used (genome? Protein?) should be indicated in the methods at least.

Further issues in the text structure limit the understanding of the first result section. The text of the first section is not organized in the logic outlined in figure 1. A reference to figure 1e seems missing and indeed the corresponding paragraph (l146-152) is placed strangely before the description of the

quality checks for the assembly. Moving this paragraph towards the end (l188?) would help the flow of this section. In addition to this, paragraph l.182-l188 is confusing, as the reference to Fig.1d is placed in a sentence that outlines a number of duplicated genes (Fig.1c) whereas Fig. 1d shows the alignment between haplotypes.

L. 197-200: Do the words "haploid complement", "haploid reference genome", "H1 complement" and "reference genome of 99KS76A-1" mean the same thing? If they do, choosing one way to refer to the sequence would clarify. If not, the difference between these needs to be stated more obviously. Perhaps due to this confusion, I do not understand what the variants added lines 207-208 are and why they need to be added separately to the total. The methods section is too sparse in this to improve the understanding either.

L.296: is the dN/dS really what is looked at here? Since there is no outgroup mentioned, wouldn't it be the pN/pS instead? If so, should the interpretation here be adapted? See 10.1016/j.tree.2006.06.015

Figure 4 has unclear axes names: what is an SV-covered ratio? It would be useful to mention in the main text what types of SV are detected here: insertions/deletions only? Copy-number variation? Inversions? Does this differ based on which method was used? This would be more relevant than mentioning a custom Python script which gives no meaningful information as to what is actually done and should be mentioned in the material and methods only.

Figure S6 is useless as it stands currently: making it bigger might help, but it is unclear to me what we are supposed to gain from it.

I am not convinced by the interpretation of the genome-wide estimation of F_{st} at lines 380-382. Wouldn't stronger purifying selection on SV lead to more SV being population-specific than synonymous SNP? I do not see that it gives any indication about SV being adaptive.

Figure 5 is in general very informative but many details in the colors make it hard to extract the information. Having consistent colors for different numbers of K would be especially useful (ex. Samples from the VI-C cluster go from orange to purple to teal to yellow) to catch at a glance the changes with increasing K. In panel C (and similar supplementary figures), the most visible color is the missing data which induces misinterpretation of the information presented. Figure S12 is very clear and it would be useful if a similar concept was implemented in figure 5 (probably not with pies as this would be too busy but perhaps colored dots for the non-admixed samples).

The text in lines 442-448 does not explicitly compare the clusters (only the identity of the admixed samples), and since the colors between Fig. S11 and 5a are different, it is hard for interested readers to do this by themselves

L.579 to 584 are unclear.

Fig. 6a bottom panel needs more information in the legend (especially as this type of plot is repeated in the supplementary figures). Do the 2 plots correspond to the ancestry of the two haplotypes? Are they two components of the same inference? What is the x-axis of the bar plot and what does the number above the bars represent?

Fig. 7b legend does not seem to correspond as these are violin plots.

Figure 8 is very useful for a general audience (such as the one targeted in this audience) which might not have all the information in mind. I actually found it so useful that I would have liked it earlier in the text. It seems to fit very well with the last paragraph of the corresponding results section (l566-571).

The protocol for the extraction of the fully assembled sample is very detailed, but this was not at all the case for the Illumina resequencing. Was the same protocol used? The short variant filtering seems extremely simple, is this really all that was done? Were there no values of calling quality used at all? Even the criteria listed seem contradictory: "minimum depth of read coverage below 3" and "minimum depth of coverage below 10" for example. How are they different? Was one used per genotype/isolate

and the other for the variant as a whole? The GATK tools (tool for the deduplication, tool for the SNP calling, etc.) and options used should also be listed here as GATK is a very complicated software, able to do many different things which could all be called "GATK pipeline" (l.833).

Data accessibility: In the copy for the reviewers, different files contained different versions of the data accessibility statement, so it is unclear which we should refer to. The main manuscript only refers to the assembly itself. Raw reads are needed for the isolate 99KS76A-1 for all methods used (Hi-C, nanopore, etc), and needed for all the Illumina resequencing (which were missing in the manuscript but mentioned in another file with a different statement).

Typos:

L. 214: A majority OF THE variants

l. 385: is continuous the right word there? Contiguous perhaps although this does not quite seem correct either?

l.450 individulas -> individuals

l. 555 Fig is repeated.

REVIEWER COMMENTS

Reviewer #1 (Remarks to the Author):

The manuscript by Guo et al. describes genome analyses of the wheat stem rust *Puccinia graminis* f. sp. *tritici*. This pathogen is a major concern to wheat production worldwide and is often used to illustrate the breakdown of effective resistance in crops (i.e. the emergence of the Ug99 lineage among stem rust populations). Genomic analyses of the species were initiated with a draft assembly by Duplessis et al (2011) and high-quality haplotype-resolved assemblies by Li et al (2019, Nat Comm). The dikaryotic lifestyle (two possibly distinct nuclei present per cell) and the repetitive genome caused challenges prior to the latest round of genome sequencing innovations. This study here adds a third high-quality genome assembly for the species and documents structural variation within the species affecting putative effector complements. The authors then focus on identified structural variant loci to genotype a broader panel of resequenced genomes. The study also expands on the work by Li et al. by finding more complete evidence for global admixture, possibly links with virulence, somatic hybridization and recombination.

Comment 1: The study tackles many different aspects of the genetic make-up of the species and explores links with virulence, as well as identifying putative admixture events. The results text could be significantly shortened so that the focus is on key independent findings. The current version is lengthy and has sections that are largely redundant or could be described more succinctly to amplify the message.

Response: Thank you for the suggestion! The manuscript was shortened to provide more concise and focused messages.

Main comments:

Comment 2: The haplotype resolution assembly strategy for the US isolate follows a well-tested pipeline following Li et al in 2019 that produced very well resolved haploid genome assemblies. The presented data shows that the new haplotype-resolved assembly is also of high quality. A minor point would be to report the BUSCO findings per haplotype in Fig 1c. My understanding is that the high number of duplicated BUSCO genes comes from having both haplotypes combined for the analyses, correct?

Response: Supplementary Table 3 is updated with BUSCO results reported per haplotype.

Comment 3: The results text in the first paragraphs would flow better though if only the essence would be kept and detailed numbers, mentions of software (incl. versions), read length, etc. only mentioned in the methods section.

Response: Thank you for the suggestion. This section was revised and most of the technical data related to genome assembly was moved to the Methods section.

Comment 4: Reporting BUSCO scores for annotated genes would be helpful to assess to the completeness of the annotation.

Response: The BUSCO scores have been added to the manuscript.

Comment 5: L197 – a more explicit statement on the global sampling directly in the results text would be helpful.

Response: The information related to sampling strategy such as geographic distribution, sampling time and representation of isolates from the previously identified clades of Pgt.

Comment 6: L218-223: I agree that the bias towards rare variants for high-impact mutations is caused most likely by purifying selection. Allele frequency spectra can be strongly affected by the demographic structure though, hence dividing the isolates into reasonably homogeneous genetic pools prior to spectra analyses would be the appropriate approach to draw broader conclusions.

Response: Population was split into more or less genetically homogeneous subsets from US and non-US, as described in the response to comment 11. The results of MAF analyses show that the trends observed in the whole population remain similar, though less statistically significant. We have added new figure (Supplementary Fig. 4) to supplementary files and also modified text of the Results section to include these analyses.

Comment 7: Linkage disequilibrium decay curves should be fitted using e.g. the model by Ingvarsson (Genetics 2015) to properly account for population genetic processes influencing r^2 . The figure panel 2d seems to show a simple smoothing function.

Response: We have used the model originally described by Remington et al., 2001 (doi: <https://doi.org/10.1073/pnas.201394398>) to fit to LD data. To reduce the effect of population subdivision, the rates of LD decay were estimated in the subsets of genetically homogeneous isolates from the US and outside of the US. The results of LD analysis are updated and presented on Fig. 2d.

Comment 8: The population genetic analyses based on π , Tajima's D and F_{st} show highly significant differences but only minute effect sizes. This important caveat is not made clearly enough in the results text.

Response: We have updated the Results section and indicated that differences in Tajima's D and F_{st} are small.

Comment 9: The property of the allele frequency spectrum expressed by Tajima's D is strongly influenced by the demographic history of a population (expansion/contraction) and subdivisions. Only in ideal and at-equilibrium populations can Tajima's D be used to identify positive versus balancing selection signatures. Looking at later analyses (i.e. Fig 5) shows that the grouping by regions produces highly heterogenous populations unsuitable for Tajima's D analyses. Please address also concerns about using Tajima's D elsewhere in the manuscript.

Response: Thank you for the suggestion. Please read response to comment 11, where we describe approaches taken to assess and reduce the effect of demography on detection of regions under selection.

Comment 10: The Tajima's D and Fst analyses would be more coherently motivated if the authors first performed unsupervised clustering (as in Fig. 5) and use the information on genetic identities to create genetically homogenous populations for selection scans / Tajima's D analyses.

Response: Thank you for the suggestion. Please read response to comment 11, where we describe approaches taken to assess and reduce the effect of demography on detection of regions under selection. We have revised this section of the manuscript by focusing on the analyses of selective sweeps following the reviewer's comment 11.

Comment 11: The XP-CLR and XP-EHH analyses are also sensitive to demographic properties. To retain such analyses in the manuscript, demographic models need to be implemented to provide a null expectation for selection signals expected from demographic factors alone. Signals in excess of such differentiation should then be used as the most likely selection signals instead of a 95th-percentile threshold. Depending on the demographic history either much less or much more loci might actually show signatures of selection. Here, carefully selecting homogeneous genetic groups is important instead of relying purely on geographic regions.

Response: Thank you to the reviewer for pointing at the possible impact of demographic factors on the results of selection scans. In the revised manuscript, we applied several approaches to reduce the potential number of falsely detected selective sweeps. 1) We used two distinct methods implemented in the XP-CLR and SweeD programs for detecting regions showing evidence of selection. Only those regions that were detected using both methods were considered under selection. 2) For analyses, we selected two samples of stem rust isolates, one from the US (15 isolates) and another outside of the US (22 isolates). Each sample of isolates was selected to be more genetically homogeneous and show no evidence of inter-sample admixture at the optimal value of $K = 6$. 3) We followed the reviewer's advice, and before conducting selection scans, we investigated distribution of selection scan test statistics in the simulated data that was generated taking into account demographic history of the two Pgt samples.

Briefly, the population size and split time history for simulations of the two Pgt samples was obtained using the methods implemented in smc++ software. The inferred population parameters were used to simulate datasets using the MaCS. For each simulated dataset diversity statistic was calculated and only those datasets that match observed data were used for downstream analyses. In total, we retained 1000 simulated datasets for selective sweep analysis. Both methods of selection scan were applied to calculate the test statistics and the maximum test statistic values from each simulation were extracted to build null distributions. The 95th percentiles of null distributions were 53.48 for XP-CLR and 4.27 for SweeD. Since these values were lower than the 95th percentiles of the test statistic distribution for XP-CLR (72.26) and SweeD (17.68) in the real-life datasets, we opted to use the empirically defined thresholds in our study.

Further, the regions shared by both scans were classified as genomic regions under selection and the number of effector-encoding genes overlapping with these regions was calculated. To assess

the chance of obtaining the set of genomic regions overlapping with effector-encoding genes (159 genomic regions), we counted the number of genomic regions in 1000 randomized datasets (mean is 80). These new results show even higher level of enrichment for effector encoding genes (~ two-fold enrichment) within the selective sweep regions than we reported in the previous version of the manuscript.

The evidence of selection was further confirmed by comparing the levels of genetic diversity, Tajima's D and population differentiation in the effector-encoding genes from the selective sweep regions with effectors located outside of the selective sweeps.

We believe that the updated approaches now used in the revised manuscript provide protection against false identification of selective sweeps and making erroneous conclusions about the effect of selection on genetic diversity of effector encoding genes.

The Methods and Results section were updated with new analyses and results. The results of analyses are shown in Fig. 3.

Comment 12: L296: dN/dS is not a ratio of non-syn. vs syn. mutations but rather a difference in rates in a phylogenetic context. The authors may mean pN/pS instead? Using dN/dS within species is controversial due to confounding effects of within-locus recombination.

Response: Thank you for detecting this error. The text was corrected.

Comment 13: L305: "Local adaptation" is not typically defined as high genetic differentiation of populations but selection of locally fit genotypes in the face of gene flow among populations thus generating GxE interactions. Please clarify the wording here depending on the actual aim of your analyses. My understanding is that you are looking for selection signals specific to individual genetic groups. Please see above for issues with using XP-CLR and the need for demographic modelling.

Response: Thank you for the clarification. These sentences were modified to provide more accurate definitions. The demographic modeling to select thresholds for statistical analysis have been performed as described in response to Comment 11.

Comment 14: L332: Does "more successful" refer to simple higher recovered numbers or challenges in recovering small vs. long SVs?

Response: This statement is not completely accurate. We found Assemblytics was not able to report some large SVs accurately. For example, it was not able to detect SV involving the *AvrSr50* locus, which has a size of ~26 kbp. This prompted us to write a custom script to extract SVs from the alignments (deposited to GitHub: https://github.com/akhunovlab/stem_rust_diversity).

Comment 15: How were the SVs retrieved from MUMmer and Assemblytics then quality filtered? Were any efforts made to identify rates of false positives or negatives by comparing

with independent approaches? Please provide some insight into the SV quality assessment and add a quality test.

Response: Discovery of SVs by whole genome comparison (assembled using long reads) is arguably the best way of detecting SVs, which is much more powerful than trying to detect SVs by aligning short or long reads to reference genomes and then trying to detect SVs based on changes in the depth of read coverage at the SV junctions or across the SV region. In our study we used whole genome comparison of 4 haplotypes assembled using long reads.

We tried to assess the accuracy of SV discovery by using long OXN reads that span entire SVs in the Pgt race RKQQC sequenced in our study. Because every OXN read is derived from a single physical molecule, we assumed that if a single OXN read spans the entire SV region, then it should be physically present in the genome. For analyses, we selected SVs discovered in RKQQC by haplotype alignments, and aligned long OXN reads to the RKQQC genome to detect reads spanning the SVs. Then, we compared SV alleles confirmed by long reads with SV alleles identified by the alignment of assembled haplotypes. The conservative estimate of concordance rate between the SVs detected by whole genome comparison and OXN read alignments was 95%. The actual concordance rate is likely higher than 95%. Due to repetitive nature of some of the SV regions, alignment of individual OXN reads was complicated and we opted to count them as failed SVs calls. The Methods and Results sections were updated.

Comment 16: The authors construct a large panel of SVs segregating among the fully assembled genomes. To overcome challenges of SV calling with short reads, k-mers were identified that should tag known SV alleles. I do not find a rigorous assessment how well this genotyping method performs. What is the likelihood that the chosen k-mers are mutated in the panel of Illumina sequenced strains? Mutated k-mers could lead to false negatives either at a certain rate. Similarly, what is the likelihood for false positive calls given the complexity of the genome and the amount of short read data processed? Assessing only the uniqueness of the k-mers in the two reference genomes seems insufficient here. Both false positive and false negative rates should be assessed given the novelty of the approach. Alternatively, I would suggest to apply and validate well tested graph-based methods (e.g. PanGenie, Giraffe, etc).

Response: Our initial efforts were indeed focused on using existing software for detecting SVs in our datasets. However, we did not find one that would detect some of the known variants that were identified by intergenomic comparison between the 99KS76A-1 and Pgt-21 genomes. This was the reason why we switched to k-mer based analyses of predefined sets of SVs detected between the sequenced Pgt genomes.

However, we do agree that the mutations in the regions covered by the diagnostic k-mers might affect the rate SV discovery. However, they will unlikely contribute to false negative rate (no variation) because they will be recorded as missing data rather than non-variable sites. To reduce the effect of variation at the sites covered by diagnostic k-mers, we filtered out those sites that had SNP variation in the Pgt panel. The accuracy SV calling for this filtered SV set was further validated by comparing the SV calls in the 99KS76A-1 genome generated by whole genome haplotype alignments and by k-mer matching approach. The high concordance rate (99.9%) between the two datasets suggests that SV genotyping strategy applied in the study is accurate.

Comment 17: L365: Please reword "accurately" as there seems to be no formal test associated with this statement.

Response: Changed to "adequately".

Comment 18: L381-382: Candidate loci for divergent selection among populations are typically identified using an outlier approach. Here, the authors seem to simply compare SNP- vs SV-based F_{st} values. This suggests that SV tend to be overall more differentiated among populations but not that SVs as a whole should be adaptive as written.

Response: The sentence was re-worded.

Comment 19: Figure 5c: I find it impossible to relate the several dozen resistance gene labels in the legend text to the appropriate column in the figure itself. Please consider splitting this information into multiple figures (or SI). Furthermore, what should the reader take from this matrix as a message?

Response: This part of the figure was removed.

Comment 20: L496: To detect admixture between populations using local ancestry, the authors based themselves on phylogenetic clades defined in previous publications. My reading of Figure 5a is that the match between these phylogenetic clades and genetic cluster assignments based on ADMIXTURE is not perfect. Therefore, I am not following why phylogenetic clades based on limited markers should better reflect genetic structure than whole-genome based ADMIXTURE analysis. My concern here is that using these weakly defined phylogenetic groups for admixture analyses can lead to false positive and false negative admixture signals. Relying on genetic clusters instead should largely solve this issue.

Response: The authors disagree with the reviewer that phylogenetic clades are not consistent with the results of Admixture. First, it could be clearly seen that with increase in K value, groups of isolates showing similar proportions of genetic ancestry assigned to K clusters tend to cluster within the sample phylogenetic clades. Second, the phylogenetic clades presented on Fig. 5a are based on genetic distances calculated using 20,000 genome-wide distributed SNPs identified in our study. This number of markers should provide robust estimates of phylogenetic relationship within the sample, and it is unlikely that the tree topology would change if the number of markers is increased further. Third, the phylogenetic clades based on 20,000 SNPs are consistent with the clade definitions made in the number of previous studies, even though they used much smaller set of SNPs. These three factors are the main reasons why in further analyses of local ancestry we used clade definitions made in the earlier quite extensive studies.

Comment 21: Figure 7 and virulence assays. What are the fine dots in panel a)? My understanding is that there are only a few dozen admixed and non-admixed individuals. Please report the statistics in full. I am unsure to understand how such a low p-value was found given a

fairly small sample size. The same applies to panels c) and d) where showing individual dots would make the analyses more transparent.

Response: We had a decent-sized panel of wheat lines carrying 47 Sr genes for phenotyping each isolate. The figure legend (now Fig. 8) is updated to provide details: “The virulence of Pgt isolates was greater in the isolates showing evidence of admixture (3 isolates x 47 Sr genes = 141 combinations) compared to the non-admixed isolates (22 isolates x 47 Sr genes = 1034 combinations)”.

Comment 22: Code availability: Please provide custom scripts mentioned in the text on a public repository.

Response: The scripts are deposited to the GitHub:
https://github.com/akhunovlab/stem_rust_diversity

Comment 23: Data availability: Where was the RNA-seq data used for the annotation deposited?

Response: This data comes from previously published studies, which were cited in the manuscript. The corresponding references are added to the Methods section “Gene prediction and functional annotation”.

Reviewer #2 (Remarks to the Author):

Guo et al. present the assembly of a haplotype resolved *Puccinia graminis* f. sp. *tritici* genome by using several sequencing data types. This new resource (a great achievement in itself) was then leveraged to identify and characterize structural variation across a Pgt panel, as well the impact of structural variations on effector gene loci. It was determined that effectors are enriched in TE-rich regions that also have a high rate of structural variation, which may lead to rapid change/modification of effector loci and aid the pathogen in evading R-gene detection. A unique aspect of this manuscript that shed new light on the emergence of novel, virulent races was the examination of local ancestry. Importantly, the somatic hybridization event that led to the generation of the Ug99 lineage was confirmed and a potential second donor lineage was identified. Overall, this manuscript was very well written and the methods/analyses used were appropriate. This work advances our understanding on how complex genome dynamics may impact host-pathogen interactions, as well as details the manner in which somatic hybridization and sexual recombination contribute to the formation of new pathogen lineages. I have outlined some suggestions and questions for further clarification for the authors below.

Comment 1: L29: change to “We utilize long-read sequencing”

Response: Corrected.

Comment 2: Supplementary Table 1: What is the connection to Supplementary Figure 12?

Response: This was a typo. Supposed to be Figure 21.

Comment 3: L125: Prior to this section, it may be helpful to briefly mention basic assembly statistics from the primary Canu assembly (or reference Supplementary Table 2 in some manner).

Response: The reference to Supplementary Table 2 is added.

Comment 4: L178-181: Are the single-copy genes located randomly throughout the genome or are there large stretches of low haplotype divergence?

Response: They were randomly distributed across the genome with an average interval of 444 kbp, and median of 295 kbp.

Comment 5: L214-215: What differentiates intergenic and upstream/downstream polymorphisms? Are upstream/downstream within a specific proximity of a gene?

Response: We used default upstream and downstream interval size (5 kb) based on snpeff manual. The legend for Fig. 2 is updated.

Comment 6: Figure 2a: Why is exon and intergenic illustrated twice on this figure panel? –

Response: Duplicated data is removed from the figure panel.

Comment 7: L221: I think this conclusion comparing nonsynonymous and synonymous SNPs may be a bit of a stretch, since the MAF distributions are very similar and not significantly different. I suggest shifting the focus of this section to the comparisons to the frequency spectrum of the higher impact mutations.

Response: Corrected.

Comment 8: L218-223: Although both stop codon and splice site mutations are considered high impact mutations, I feel it may be more relevant to compare these categories separately.

Response: These two classes of mutations lead in most cases to the same outcome – premature termination codon. In that sense, these are two equivalent mutation types. In most diversity studies these two mutation types are analyzed jointly.

Per your suggestion, we tried to analyze these two classes separately. The results are shown below. They show the same trend (both classes of mutations tend to be overrepresented among rare variants). However, due to reduction in sample size, only stop codon vs. syn mutations remained significant in Kolmogorov-Smirnov test with p-value = 0.02. In splice-site disruption vs. syn comparison, p-value increased to p = 0.1. Since these results are due to changes in sample sizes and unlikely have much biological significance, we would prefer to analyze these two classes of mutations jointly.

Comment 9: L240-243: As the authors alluded to here, the existence of population structure can influence the calculation of LD. I suggest calculating LD decay for each population separately (for those with sufficient representation), in addition to the total population presented here, to obtain a better estimate.

Response: Performed as suggested. We used two populations: 1) combining Pgt isolates from Europe/Africa/Asia and 2) including Pgt isolates from US to model the rate of LD decay. For details, please see response to Comment 7 from Reviewer 1.

Comment 10: L248: change to “Tajima’s D”

Response: This section has been re-written and typo is corrected.

Comment 11: L312: Are these 14,436 outliers windows or combined adjacent genomic regions?

Response: This section was re-written to use overlap between the two selection scan methods, XP-CLR and SweeD, to claim that region is showing the evidence of selection. In all cases, neighboring regions showing evidence of selection are merged, as was mentioned in the Methods section.

Comment 12: L314: The number of effector genes overlapping with randomly sampled genomic regions was still unclear to me. I suggest explicitly stating this number to clarify the comparison to the 818 effector genes that overlap outlier regions.

Response: This section was revised following the Reviewer 1 suggestions using new approaches of selection scans with indication of overlap with the random dataset: “We identified 1,602 genomic outlier regions shared by both the XP-CLR and SweeD scans, and among them 159 genomic regions overlapping with 120 effectors. This overlap is nearly two times higher than the average randomly selected genomic regions of equal size overlapping with effectors (80) (Fig. 3c).”

We compared number of regions overlapping with effectors rather than number of effectors, because results based on number of effectors are not accurate, since effectors length varies and

clustered selective sweep regions could share same effector. In order to further validate selective sweep regions are enriched with effectors, we compared the total size of selective sweep regions overlapping with effectors (36,189), which was exceeding two times than random selected regions overlapping with effectors (mean of 16,671).

Comment 13: L319: To what degree were the results from the XP-CLR and XP-EHH tests correlated? Did some of the same conclusions drawn from the XP-CLR test hold true for the XP-EHH test (particularly the enrichment of effectors)? Do these tests differ in their abilities to detect hard vs soft sweeps?

Response: We have substantially revised this section of the manuscript. For details, please refer to the response to Comment 11 from Reviewer 1. Based on the reviewers' suggestions, we 1) selected two Pgt samples that show low levels of inter-sample admixture, 2) modeled the Pgt samples' demography to investigate the null distribution of selection scan test statistic for XP-CLR and SweeD, and select thresholds for identifying outlier windows, 3) used overlap between the two selection scans, XP-CLR and SweeD, to claim that a region is under selection. The level of effector enrichment in the revised analyses are much higher than detected in the previous version of the manuscript. The Results and Methods sections have been updated.

Comment 14: L305-327: Although it was nice to highlight previously validated effectors identified in these selection analyses, I think it would be informative to elaborate further on other (presently unvalidated) effector loci that these tests detected. Were there any candidate effectors that showed even stronger evidence for positive selection than AvrSr35 or AvrSr50?

Response: We provided the list of 120 effectors in Supplementary Table 13 that show evidence of selection in the Pgt panel.

Comment 15: L383: Here it is stated that 560 SVs overlapped with 707 predicted effector genes, but in L338, it is stated that 806 effectors were affected by SVs. Is the former estimate from the Pgt panel and k-mer analysis, and the latter from just the comparison of 99KS76A-1 and Pgt21-0? Please clarify.

Response: We have updated results. Now we report all SVs that overlap with 947 effector encoding genes.

“In the Pgt panel, 890 SVs overlap with the 947 predicted effector-encoding genes (Supplementary Data 4).”

Comment 16: L396-400: Is the prevalence of Sr35 and Sr50 resistance genes in local cultivars known for these regions? If so, it would be interesting to discuss this in regards to these results.

Response: Both genes are introgression from wild relatives and have not been broadly used until Ug99 race outbreak. At the time points of Pgt sampling there were no broadly grown cultivars that carried any of these genes.

Comment 17: L450: change to “individuals”

Response: Corrected.

Comment 18: L819: How was de novo transcript assembly conducted?

Response: We used our previously developed assemblies published in Salcedo, A. et al. Science 358, 1604–1606 (2017). We clarified in the sentence that transcripts were taken from this study.

Reviewer #3 (Remarks to the Author):

In general, I enjoyed reading this manuscript. It is well written, the conclusions are supported by the analyses, and most results are clear from the text and figures. I believe that the community will find these results of interest, as they build from previous research and add more insights into the genomic evolution of an important pathogen. I nevertheless have some comments mostly related to clarity improvement that should be addressed in revisions.

Comment 1: The introduction contains the needed information to understand the questions, context, and results overall. However, the structure of the text makes it hard to really understand it, the paragraphs alternate between information that is very specific to the species studied (ex.: paragraphs 1 and 3) and generalities about fungal pathogen genomes (ex.: paragraphs 2 and 4). Reorganizing the paragraphs to improve the flow would help the reader.

Response: Thank for the suggestion! Introduction was re-arranged to present information from more general to more specific.

Comment 2: In several cases, method information is presented outside of the methods section in such detail that it hinders the general understanding of the workflow used. This is already the case in the last paragraph of the introduction: it is not necessary at this point in the manuscript to know that SV was detected specifically with kmers of 31 bp, for example. The first section of the results contains text that is almost word for word the content of the corresponding material and methods (from l.120 to 145). At this point in the text, the reader needs more general information and less technical details. As an example, what are Hi-C reads and how are they helpful in assembling genomes is more relevant than the specific version of the software used to analyze them, which is clearly at its place in the methods.

Response: Thank you for the suggestion. We moved redundant description of methods (assembly, k-mer analysis etc.) into the Methods section.

Comment 3: Some points of the process are not very clear either: line 134/135 it seems that the Pgt21-0 assembly was used to gather contigs into groups. How meaningful then is the collinearity observed in line 146 between the two assemblies if the one was indeed used to build the other? Which BUSCO mode was used should be indicated in the methods at least.

Response: The ordering of contigs was based on Hi-C data and no ordering information from Pgt21 assembly was used. Comparison with Pg21 genome was only used to identify pairs of contigs that are allelic (from distinct nuclei) and also assign assembled contigs into previously defined 18 homologous groups from the Pgt21 assembly. We changed wording in the Methods section to clarify these points: “The contigs representing allelic variants from distinct nuclei were identified by aligning gene models from the Canu assembly to the assembled chromosomes of the Pgt21-0 isolate. This information was also used to group contigs into 18 homologous groups (HGs).”

We run BUSCO in the genome mode, which is now indicated in the Methods section.

Comment 4: Further issues in the text structure limit the understanding of the first result section. The text of the first section is not organized in the logic outlined in figure 1. A reference to figure 1e seems missing and indeed the corresponding paragraph (1146-152) is placed strangely before the description of the quality checks for the assembly. Moving this paragraph towards the end (1188?) would help the flow of this section. In addition to this, paragraph 1.182-1188 is confusing, as the reference to Fig. 1d is placed in a sentence that outlines a number of duplicated genes (Fig. 1c) whereas Fig. 1d shows the alignment between haplotypes.

Response: Thank you for noticing this error. We have rearranged Fig. 1 panels to follow the logic of data presentation in the text.

Comment 5: L. 197-200: Do the words “haploid complement”, “haploid reference genome”, “H1 complement” and “reference genome of 99KS76A-1” mean the same thing? If they do, choosing one way to refer to the sequence would clarify. If not, the difference between these needs to be stated more obviously. Perhaps due to this confusion, I do not understand what the variants added lines 207-208 are and why they need to be added separately to the total. The methods section is too sparse in this to improve the understanding either.

Response: Thank you for noticing inconsistency in the usage of terminology. H1 haplotype is a combination of sequences from haplotypes E and F that are unique. Both haplotypes have been used separately as references and then SNPs discovered in one haplotype but not in the other were merged. We revised this section for clarity.

Comment 6: L.296: is the dN/dS really what is looked at here? Since there is no outgroup mentioned, wouldn't it be the pN/pS instead? If so, should the interpretation here be adapted? See 10.1016/j.tree.2006.06.015

Response: Changed to pN/pS.

Comment 7: Figure 4 has unclear axes names: what is an SV-covered ratio? It would be useful to mention in the main text what types of SV are detected here: insertions/deletions only? Copy-number variation? Inversions? Does this differ based on which method was used? This would be more relevant than mentioning a custom Python script which gives no meaningful information as to what is actually done and should be mentioned in the material and methods only.

Response: The legend for Figure 4 was modified. The axes show the proportion of 1 Mb window occupied by TEs and SVs calculated for non-overlapping windows spanning entire reference genome. The types of SVs are listed in the manuscript now (deletions, insertions, contractions, and expansions). We indicated in Methods that all four types of mutations were discovered using both methods: “There are four types of SVs were detected using both methods: insertions, deletions, expansions and contractions (Supplementary Fig. 4).”.

Comment 8: Figure S6 is useless as it stands currently: making it bigger might help, but it is unclear to me what we are supposed to gain from it.

Response: Figure S6 was removed.

Comment 9: I am not convinced by the interpretation of the genome-wide estimation of F_{st} at lines 380-382. Wouldn't stronger purifying selection on SV lead to more SV being population-specific than synonymous SNP? I do not see that it gives any indication about SV being adaptive.

Response: We agree that these results are not sufficient to claim that SVs are adaptive. We modified language in the results. But it is expected that the accumulation of private SVs in one of the populations would lead to stronger genetic differentiation.

Comment 10: Figure 5 is in general very informative but many details in the colors make it hard to extract the information. Having consistent colors for different numbers of K would be especially useful (ex. Samples from the VI-C cluster go from orange to purple to teal to yellow) to catch at a glance the changes with increasing K . In panel C (and similar supplementary figures), the most visible color is the missing data which induces misinterpretation of the information presented. Figure S12 is very clear and it would be useful if a similar concept was implemented in figure 5 (probably not with pies as this would be too busy but perhaps colored dots for the non-admixed samples).

Response: Though coloring in the population structure analyses does help with data interpretation, it has limited meaning across different values of K . We tried our best using available software (pophelper) to control color-coding of different clusters. While at lower values of K the software provides consistent color-coding, at $K = 5$ and above the color coding becomes less consistent.

Figure 5 was updated and simplified to retain only relevant information for discussing population structure and admixture.

Figure S12 as well as the results of SplitsTree analyses were removed. These analyses were used to demonstrate existence of admixture in the Pgt panel, and essentially demonstrate the same trends observed in Admixture. To shorten and write more concise story, we decided to remove them.

Comment 11: The text in lines 442-448 does not explicitly compare the clusters (only the identity of the admixed samples), and since the colors between Fig. S11 and 5a are different, it is hard for interested readers to do this by themselves

Response: We changed color-coding in this supplementary figure to match clusters colors (K=4) with those in Fig 5b (K=5).

Comment 12: L.579 to 584 are unclear.

Response: We tried to revise. The main idea is simple. We want to group isolates into 2 groups: admixed and non-admixed. We had important requirement: for each portion of ancestry in K clusters in the isolates from the admixed group, we should have an isolate in the non-admixed group with at least 95% of its ancestry assigned to the same cluster. Let's take example when admixed isolate has 20% of ancestry assigned to cluster 1 and 80% of ancestry assigned to cluster 2. Then in non-admixed group we should have at least one isolate that has more than 95% of ancestry assigned to cluster 1 and at least one isolate that has more than 95% of ancestry assigned to cluster 2.

Comment 13: Fig. 6a bottom panel needs more information in the legend (especially as this type of plot is repeated in the supplementary figures). Do the 2 plots correspond to the ancestry of the two haplotypes? Are they two components of the same inference? What is the x-axis of the bar plot and what does the number above the bars represent?

Response: The legend for Fig. 6a was updated. The bottom panel's x-axis shows the copying probabilities of two ancestries from isolate 4A and 7A. The split of proportions of ancestry in two-way admixture model for the panel of three isolates is shown on the top (0.608 and 0.392).

Comment 14: Fig. 7b legend does not seem to correspond as these are violin plots.

Response: The legend is corrected.

Comment 15: Figure 8 is very useful for a general audience (such as the one targeted in this audience) which might not have all the information in mind. I actually found it so useful that I would have liked it earlier in the text. It seems to fit very well with the last paragraph of the corresponding results section (1566-571).

Response: The figure was moved and relabeled as Fig. 7.

Comment 16: The protocol for the extraction of the fully assembled sample is very detailed, but this was not at all the case for the Illumina resequencing. Was the same protocol used? The short variant filtering seems extremely simple, is this really all that was done? Were there no values of calling quality used at all? Even the criteria listed seem contradictory: "minimum depth of read coverage below 3" and "minimum depth of coverage below 10" for example. How are they different? Was one used per genotype/isolate and the other for the variant as a whole? The GATK tools (tool for the deduplication, tool for the SNP calling, etc.) and options used should

also be listed here as GATK is a very complicated software, able to do many different things which could all be called “GATK pipeline” (l.833).

Response: The methods of DNA isolation for long-read sequencing are critical, because no good data could be generated using poorly extracted low-molecular weight DNA. This is the reason for providing detailed description of the protocol. The method of DNA extraction for short read re-sequencing followed the same one described by Olivera et al., 2015. This reference to this extraction method was added to the Methods section.

There is no contradiction with the filtering criteria. It appears that the issue is in unclear writing, which led to misunderstanding. All variants were filtered to remove sites with total or alternative allele coverage below 3. But, for sites that are singletons in the population (minor allele count = 1), more strict criteria was applied for depth of coverage (less than 10). While for non-rare variants, multi-sample calling could pull information across samples to improve calling accuracy even at the low depth of read coverage, for rare allele the only way to improve calling accuracy is to increase the depth of read coverage. The concordance of 99% (reported in the Results section) between genotype calls generated for 99KS76A-1 using short read with genome assembly suggest that our approach generated reliable SNP dataset. We updated the Methods section to clarify how filtering was performed.

We are aware that GATK is complex program. However, many defaults setting of GATK selected over the last decade of research provide quite reliable variant calls. We have added additional details, such as that we used HaplotypeCaller option of GATK. We emphasized in the Methods section that we used default settings of HaplotypeCaller GATK for variant calling.

Comment 17: Data accessibility: In the copy for the reviewers, different files contained different versions of the data accessibility statement, so it is unclear which we should refer to. The main manuscript only refers to the assembly itself. Raw reads are needed for the isolate 99KS76A-1 for all methods used (Hi-C, nanopore, etc), and needed for all the Illumina resequencing (which were missing in the manuscript but mentioned in another file with a different statement).

Response:

All sequence files for 99KS76A-1 genome assembly, including nanopore, pacbio, Hi-C and Miseq, have been deposited to NCBI under same BioProject with assembly (PRJNA313186), and will be released either early next year or upon publication of this manuscript, which ever will come first.

Illumina sequence files for the whole Pgt panel have been deposited to NCBI under BioProject PRJNA803546, and will be released either early next year or upon publication of this manuscript, which ever will come first.

All these information have been added to Data accessibility in the manuscript.

Comment 18: Typos:

L. 214: A majority OF THE variants

l. 385: is continuous the right word there? Contiguous perhaps although this does not quite seem correct either?

1.450 individulas -> individuals

1. 555 Fig is repeated.

Response: Thank you! Corrected.

Reviewers' Comments:

Reviewer #1:

Remarks to the Author:

The revised manuscript by Guo et al. is much improved in many aspects. I paid particular attention to my own previous comments and I am well satisfied with the revisions. I think the reduced length of the results, the proper subdivision into genetic groups, the modelling and some new randomization tests create a much more robust message.

Some minor points:

- l63: "start" necessary?
- l146: which **is** consistent
- l213: patterns in **the** genome. Or rather "populations"?
- l217 and onwards: Reduced recombination could indeed contribute to slower LD decay, however population bottlenecks (hence, effective population size) are often the key factor in explaining such differences.
- l238: remove "the"
- l507: "shown with"
- l812: have **a** significant
- l813: **a** more

Reviewer #2:

Remarks to the Author:

The authors have done a great job at responding to all of the previous comments/suggestions. Overall, the paper reads well and most previous concerns were sufficiently addressed. I only have one additional comment/suggestion after reading the revised manuscript.

L526: After re-reading this, I feel that the conclusion that admixture between lineages is linked to increased virulence is possibly overstated. Although there are statistically significant differences, the effect appears to be rather small and the distributions don't appear strikingly different, considering the sample size. Additionally, I do think the small sample size used for the admixed isolates (n=3) might be too low to make broad conclusions about admixed isolates in general. Is it possible that these three isolates are slightly more virulent than the median of the non-admixed population by random chance? I suggest to at least address this in the text and make the magnitude of these differences a little more clear.

Reviewer #3:

Remarks to the Author:

I am satisfied that the changes made to the manuscript answer my comments about the initial submission. The revised manuscript is shortened with a more to-the-point message, the figures are improved where it was needed and the issues with the methods were addressed and corrected.

Reviewer #1 (Remarks to the Author):

Minor comment:

- 163: "start" necessary?

Response: Removed.

- 1146: which **is** consistent

Response: Corrected

- 1213: patterns in **the** genome. Or rather "populations"?

Response: changed to "populations"

- 1217 and onwards: Reduced recombination could indeed contribute to slower LD decay, however population bottlenecks (hence, effective population size) are often the key factor in explaining such differences.

Response: The sentence was modified: "Among factors explaining such differences in LD decay could be either changes in effective population size due to population bottleneck or reduced frequency of inter-lineage recombination...."

- 1238: remove "the"

Response: Corrected

- 1507: "shown with"

Response: Corrected

- 1812: have **a** significant

Response: Corrected

- 1813: **a** more

Response: Corrected

Reviewer #2 (Remarks to the Author):

Comment 1: L526: After re-reading this, I feel that the conclusion that admixture between lineages is linked to increased virulence is possibly overstated. Although there are statistically significant differences, the effect appears to be rather small and the distributions don't appear strikingly different, considering the sample size. Additionally, I do think the small sample size used for the admixed isolates (n=3) might be too low to make broad conclusions about admixed isolates in general. Is it possible that these three isolates are slightly more virulent than the median of the non-admixed population by random chance? I suggest to at least address this in the text and make the magnitude of these differences a little more clear.

Response: After some consideration, we have decided to apply different approach to analyzing disease resistance scores of admixed and non-admixed Pgt isolates. This approach is more conservative and straightforward. We have used MOSAIC-based assignment of Pgt isolates into admixed and ancestral clades (shown in Figure 7). Then, for each trio of clades (1 admixed and 2 source clades), we have identified Sr genes that were effective against all Pgt isolates from both SOURCE clades. Within these Sr genes we counted those that became ineffective in at least half of the isolates from the ADMIXED clade. In these analyses, we have used only those clades that had phenotypes for at least three samples. We argue that the loss of efficiency for these Sr genes could have occurred only after admixture and could be associated with the stepwise accumulation of mutations in Pgt clade and/or admixture event. If we take an example of recently originated Clade I (Ug99 – detected only in 1998), out of analyzed 13 Sr genes that are effective against all isolates from two source clades IX and II, 4 Sr genes (30%) lost their efficiency against all Ug99 group isolates. This loss of resistance against all Ug99 lineage isolates is most likely linked with admixture. Using this approach, we show that on average 15% of Sr genes could have lost their efficiency after admixture event.

We have updated Results section to present this analysis (section “Relationship between admixed origin and virulence in the Pgt panel”). New supplementary table is added (Table S15). The panel 8a was removed from figure 8.